# JNK-dependent cell cycle stalling in G2 promotes survival and senescence-like phenotypes in tissue stress

Andrea Cosolo[1,2], Janhvi Jaiswal[3], Gábor Csordás[4,5], Isabelle Grass[2,6,7], Mirka Uhlirova[4,5,8], Anne-Kathrin Classen[2,6,7]*

[1]Center for Biological Systems Analysis, University of Freiburg, Freiburg, Germany; [2]Faculty of Biology, Ludwig-Maximilians-University Munich, Munich, Germany; [3]Spemann Graduate School of Biology and Medicine (SGBM), University of Freiburg, Freiburg, Germany; [4]Cologne Excellence Cluster on Cellular Stress Responses in Aging-Associated Diseases (CECAD), University of Cologne, Cologne, Germany; [5]Institute for Genetics, University of Cologne, Cologne, Germany; [6]Centre for Biological Signalling Studies (BIOSS), University of Freiburg, Freiburg, Germany; [7]Centre for Integrative Biological Signalling Studies (CIBSS), University of Freiburg, Freiburg, Germany; [8]Center for Molecular Medicine Cologne, University of Cologne, Cologne, Germany

**Abstract** The restoration of homeostasis after tissue damage relies on proper spatial-temporal control of damage-induced apoptosis and compensatory proliferation. In *Drosophila* imaginal discs these processes are coordinated by the stress response pathway JNK. We demonstrate that JNK signaling induces a dose-dependent extension of G2 in tissue damage and tumors, resulting in either transient stalling or a prolonged but reversible cell cycle arrest. G2-stalling is mediated by downregulation of the G2/M-specific phosphatase String(Stg)/Cdc25. Ectopic expression of *stg* is sufficient to suppress G2-stalling and reveals roles for stalling in survival, proliferation and paracrine signaling. G2-stalling protects cells from JNK-induced apoptosis, but under chronic conditions, reduces proliferative potential of JNK-signaling cells while promoting non-autonomous proliferation. Thus, transient cell cycle stalling in G2 has key roles in wound healing but becomes detrimental upon chronic JNK overstimulation, with important implications for chronic wound healing pathologies or tumorigenic transformation.
DOI: https://doi.org/10.7554/eLife.41036.001

*For correspondence:
anne.classen@zbsa.uni-freiburg.de

Competing interests: The authors declare that no competing interests exist.

## Introduction

Stress signaling pathways activated by tissue injury set up precise spatio-temporal patterns of apoptosis, proliferation and survival that are required to repair a tissue (*Fuchs and Steller, 2015*; *Sun and Irvine, 2014*). Accordingly, deregulation of injury-induced signals and cell behaviors are associated with ageing or pathologies, such as non-healing chronic wounds and cancer (*Martin and Nunan, 2015*; *Neves et al., 2015*; *Taniguchi and Karin, 2018*). However, how injury-induced signals precisely balance proliferation and apoptosis to restore a tissue of the correct size is still not fully understood.

Cell proliferation during tissue repair is regulated by controlling cell cycle progression, promoting re-entry of quiescent cells into the cell cycle or accelerating cell division rates. In contrast to acceleration, tissue stress can also induce cell cycle arrest. Excess cellular damage or deregulated signaling environments cause cells to arrest in G0 and enter senescence - a state characterized by resistance

to apoptosis and senescence-associated secretory phenotypes (SASP) (*Hernandez-Segura et al., 2018*; *Neves et al., 2015*; *Pluquet et al., 2015*; *Salama et al., 2014*). SASP is linked to persistent production of signaling molecules, secretion of ECM degrading enzymes, as well as an upregulation of autophagy, unfolded protein response (UPR), ROS and an increase in cell size (*Hernandez-Segura et al., 2018*; *Neves et al., 2015*; *Pluquet et al., 2015*; *Salama et al., 2014*). Strikingly, recent studies suggest that senescent cells are required at wound sites to promote wound closure and cell plasticity (*Demaria et al., 2014*; *Ritschka et al., 2017*) and in development to promote morphogenesis of embryonic structures (*Davaapil et al., 2017*; *Muñoz-Espín et al., 2013*). Thus, while cell cycle arrest and senescence are often considered to be an aberrant by-product of stress responses, it emerges that arrested cells interface in a little appreciated way with physiological events during tissue regeneration.

*Drosophila* imaginal discs (*Figure 1—figure supplement 1A*) have provided deep insights into stress signals and responses to tissue injury. The JNK/MAPK-cascade is among the earliest pathways activated by physical wounding (*Bosch et al., 2005*; *Rämet et al., 2002*), loss of epithelial polarity (*Igaki, 2009*; *Igaki, 2009*) or apoptosis (*Ryoo et al., 2004*; *Shlevkov and Morata, 2012*). JNK activates multiple transcription factors, such as AP-1 (*Eferl and Wagner, 2003*; *Külshammer et al., 2015*), and is required for wound closure (*Bosch et al., 2005*; *Ríos-Barrera and Riesgo-Escovar, 2013*), elimination of damaged cells (*Chen, 2012*; *Moreno et al., 2002*; *Shlevkov and Morata, 2012*) and compensatory proliferation replacing lost tissues (*Bergantiños et al., 2010*; *Bosch et al., 2008*; *Ryoo et al., 2004*; *Sun and Irvine, 2014*). Feed-back loops acting through ROS, p53 and the initiator caspase Dronc maintain JNK activity until tissue homeostasis is restored (*Brock et al., 2017*; *Khan et al., 2017*; *Shlevkov and Morata, 2012*; *Wells et al., 2006*). However, how JNK signaling is balanced to eliminate damaged cells and to promote compensatory proliferation is little understood.

Apoptotic cells stimulate compensatory proliferation of the surrounding tissue by JNK-dependent activation of growth and survival pathways including Hippo/Yorkie and JAK/STAT (*Fuchs and Steller, 2015*; *Pastor-Pareja and Xu, 2013*; *Sun and Irvine, 2011*; *Zielke et al., 2014*). Importantly, preventing execution of apoptosis in damaged, aberrant or tumorigenic cells causes chronic signaling and non-autonomous overgrowth in fly tissues (*Fuchs and Steller, 2015*; *Herz et al., 2006*; *Martín et al., 2009*; *Pastor-Pareja and Xu, 2013*; *Pérez-Garijo et al., 2004*; *Pérez-Garijo et al., 2009*; *Ryoo et al., 2004*; *Uhlirova et al., 2005*). However, which autonomous and non-autonomous mechanisms drive compensatory proliferation remains to be fully elucidated.

We employ surgical injury of wing imaginal discs (*Bryant, 1971*; *Yoo et al., 2016*) and cell ablation induced by pro-apoptotic transgenes (*Herrera et al., 2013*; *Smith-Bolton et al., 2009*) to study how injury-induced JNK signaling, compensatory proliferation and survival unexpectedly link to control of cell cycle progression. While stress-induced cell cycle arrest and senescence in flies are little understood (*Nakamura et al., 2014*; *Wells et al., 2006*), we propose that JNK-induced G2 stalling exhibits senescence-like qualities in *Drosophila*.

## Results

### Tissue injury induces a transient G2-shift

To investigate how imaginal disc regeneration may be regulated by cell cycle progression, we induced acute surgical injury of wing imaginal discs in situ. Consistent with previous reports (*Bosch et al., 2005*; *Mattila et al., 2005*; *Rämet et al., 2002*), we observed activation of the JNK reporter *puc*-LacZ near the wound site 6 hr post-injury (*Figure 1A–B*). Strikingly, upregulation of the JNK reporter coincided with a pronounced shift of cells towards a G2-dominated cell cycle profile visualized by the G2-specific FUCCI reporter mRFP-NLS-CycB[1-266] (*Figure 1A',B'*; *Figure 1—figure supplement 1B,C*) (*Zielke et al., 2014*). We confirmed this finding by flow cytometry, where cells from injured imaginal discs positive for the alternative JNK-reporter *TRE*-RFP (*Chatterjee and Bohmann, 2012*) exhibited a pronounced G2-shift (*Figure 1D,E*). This response could be narrowed down to the injured pouch domain by flow cytometry analysis of *rotund(rn)-GAL4, UAS-GFP* expressing cells, which normally have little *TRE*-activity (*Figure 1—figure supplement 1D-E'*). Importantly, injury-induced JNK-activity and the G2-profile were transient events, as both decreased by 16 hr post-injury (*Figure 1C,C'*). These observations suggest that JNK activity correlates with a G2 cell

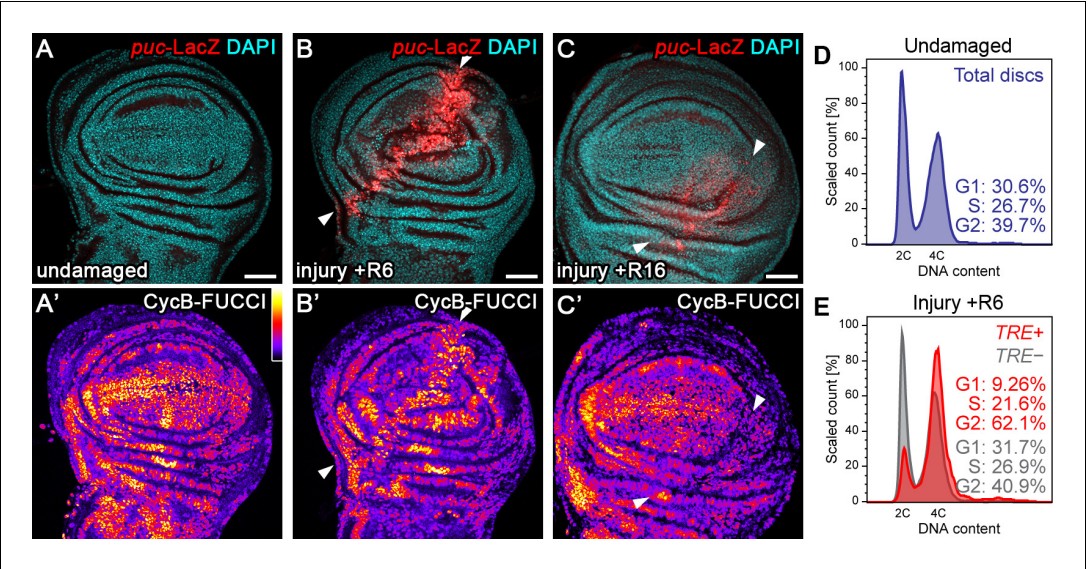

**Figure 1.** Tissue injury induces a transient G2-shift. (A–C') Undamaged control wing disc (A), a wing disc with surgical damage 6 hr (B) or 16 hr (C) into the recovery (R) period. Wing discs were counterstained with DAPI (cyan in A-C) and express the JNK-reporter *puc*-LacZ (red in A-C) as well as the G2-specific FUCCI reporter *ubi-mRFP-NLS-CycB*^1-266 (see *Figure 1—figure supplement 1B,C*) visualized using a thermal LUT (A'–C'). Arrows indicate injury axis (B,C). A quantification of JNK reporter (*TRE*-RFP) activity over time is presented in *Figure 5M*. (D–E) Flow cytometry analysis of DNA content in undamaged control wing discs (D) and wing discs with surgical damage 6 hr into the recovery period (E). JNK-signaling cells in damaged discs were detected by activation of *TRE*-RFP. *TRE*-RFP positive cells in undamaged control discs represent only a 2.5% of the total cell population and are thus not separately visualized. Detected events were plotted as counts scaled to mode against fluorescence intensity of the DNA stain Hoechst. Scale bars: 50 μm.

DOI: https://doi.org/10.7554/eLife.41036.002

The following figure supplement is available for figure 1:

**Figure supplement 1.** Tissue injury induces a transient G2-shift.

DOI: https://doi.org/10.7554/eLife.41036.003

cycle profile and that this correlation may be a transient component of physiological wound healing processes.

## Stress-dependent JNK activity correlates with G2-stalling

To more quantitatively investigate this cell cycle shift, we turned to models of tissue injury and regeneration based on targeted expression of pro-apoptotic transgenes. We first induced expression of *TNFα/eiger* (*egr*) under the control of *rnGAL4* on developmental day 7, and limited expression to 24 hr by a temperature-sensitive GAL80-repressor (*rn^ts>*) (*Figure 2—figure supplement 1A*). As described previously, we observed extensive cell death resulting in reduced adult wing sizes (*Herrera et al., 2013*; *La Fortezza et al., 2016*; *Smith-Bolton et al., 2009*) and broad activation of the JNK-reporter *TRE*-RFP in and around *egr*-expressing cells (*Figure 2—figure supplement 1B,C*) (*La Fortezza et al., 2016*). Importantly, FUCCI assays revealed a pronounced G2-shift of cells at the center of *egr*-expressing domains (*Figure 2A,B*). Flow cytometry analysis confirmed that cells positive for *TRE*-RFP (*Figure 2C,C'*), and particularly the GFP-labeled lineage of *egr*-expressing cells (*Figure 2—figure supplement 1B',C'*), exhibit a pronounced G2-profile. The marked cell cycle changes prompted us to investigate if JNK-signaling cells were actively cycling. EdU incorporation assays (*Figure 2A,B,D*) and staining for phospho-Histone 3 (pH3) (*Figure 2E,F*) revealed that DNA replication activity and mitotic cells were absent from G2-shifted, *TRE*-positive domains in *egr*-expressing discs. Of note, *TRE*-positive G2 cells were larger in size than G2 cells from undamaged control discs (*Figure 2—figure supplement 1D*). Combined, these observations confirm a pronounced correlation between injury-induced JNK activity and a G2-dominated cell cycle observed in surgically injured discs and suggest that the G2 profile represents a cell cycle arrest.

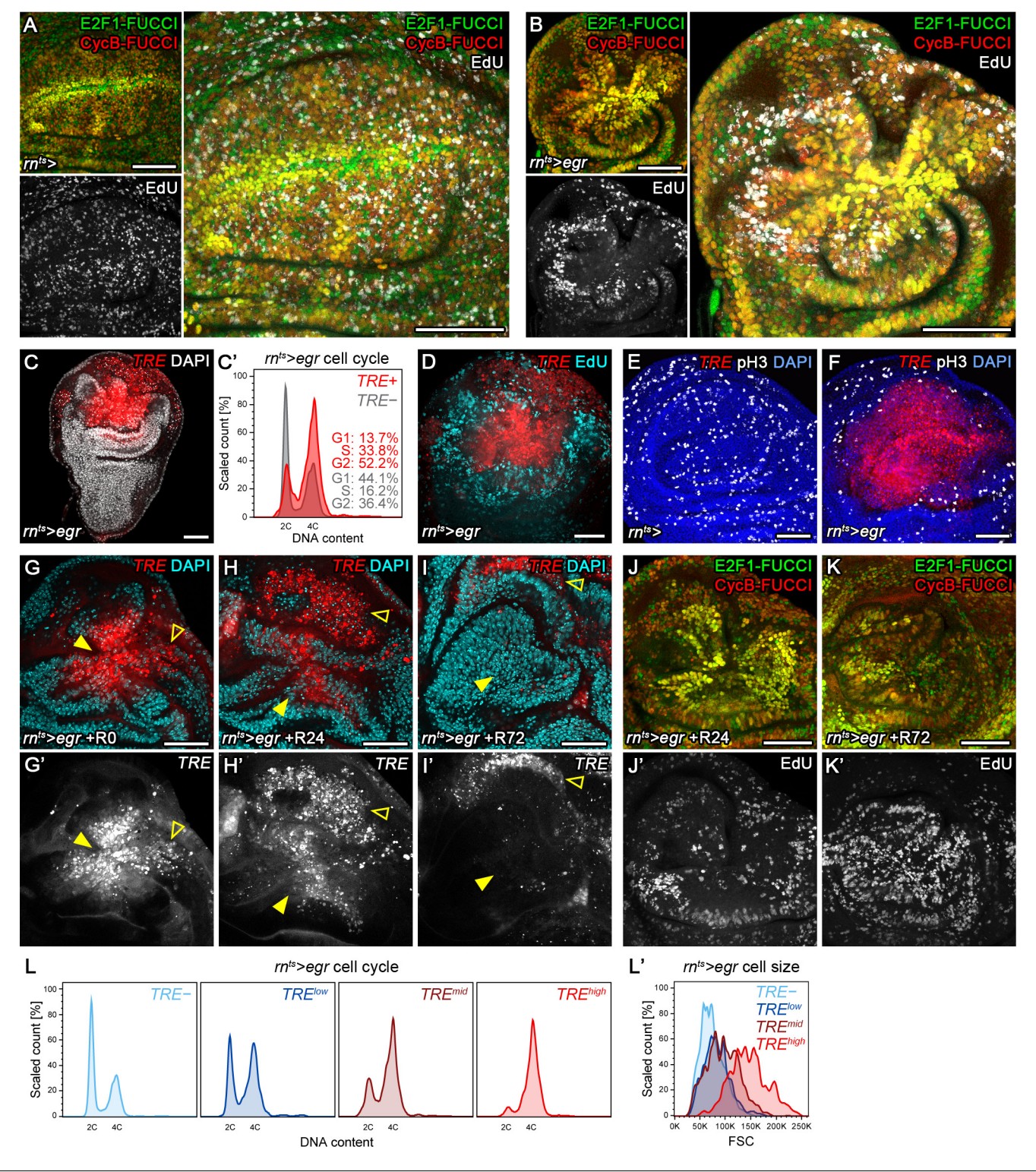

**Figure 2.** Stress-induced JNK activity correlates with G2-stalling. (**A,B**) Control wing disc (**A**) and a wing disc at 0 hr into the recovery period, after 24 hr of *egr*-expression in the pouch domain (**B**) (see ***Figure 2—figure supplement 1A***). Discs also express the complete FUCCI reporter system consisting of *ubi*-mRFP-NLS-CycB$^{1-266}$ (red) and *ubi*-GFP-E2f1$^{1-230}$ (green) and were analyzed for EdU incorporation (grey) to reveal DNA replication activity. The field of view includes the pouch and hinge domain of the disc. The horizontal G1 and G2 pattern in control discs (**A**) represents normal developmental pattern at the dorsal-ventral compartment boundary. Note the intensely labeled G2-cells lacking EdU incorporation activity at the center of the folded pouch tissue in the *egr*-expressing disc (**B**). (**C**) JNK-signaling cells in *egr*-expressing discs were detected by activation of *TRE*-RFP (red). Discs were

*Figure 2 continued on next page*

*Figure 2 continued*

counterstained with DAPI (gray). (C') Flow cytometry analysis of DNA content in *TRE*-positive (red) and *TRE*-negative (gray) cells from *egr*-expressing discs. Detected events were plotted as counts scaled to mode against fluorescence intensity of the DNA stain Hoechst. (D–F) Control wing disc (E) and *egr*-expressing discs at 0 hr into the recovery period (D,F) expressing the JNK-reporter *TRE*-RFP (red). Discs were assessed for cell cycle activity by EdU incorporation to reveal DNA replication (cyan in D) and by staining for phospho-H3 to reveal mitotic cells (pH3) (gray in E,F). Note pronounced lack of either in JNK-signaling domains. (G–K') Formerly *egr*-expressing discs at 0 hr (R0), 24 hr (R24) and at 72 hr (R72) into the recovery period. Discs were counterstained with DAPI (cyan in G-I) and either express the JNK-reporter *TRE*-RFP (G'-I', red in G-I) or the FUCCI reporters (J,K). FUCCI-reporters expressing discs were assayed for EdU incorporation to reveal DNA replication (J',K'). Compare J-K' to B. A quantification of TRE reporter activity over time is presented in *Figure 5N*. Filled arrows point to the pouch domain where formerly *egr*-expressing cells and the regenerating tissue is located. Open arrows point to apoptotic debris. (L,L') Flow cytometry analysis of *TRE*-RFP reporter activity, DNA content (Hoechst) and cell size (forward scatter, FSC). *TRE* reporter activity was divided into bins of RFP fluorescence intensity. Cells from four bins (negative, low, medium and high RFP intensity) were represented by different shades and plotted for their DNA content and cell size. Note that cells in the high *TRE* bin are almost exclusively in G2 and are the largest in size. Maximum projections of multiple confocal sections are shown in A,B,D-F,J-K'. Scale bars: 50 µm.

DOI: https://doi.org/10.7554/eLife.41036.004

The following figure supplement is available for figure 2:

**Figure supplement 1.** Stress-induced JNK activity correlates with G2-stalling.

DOI: https://doi.org/10.7554/eLife.41036.005

We wanted to investigate the fate of G2-arrested cells and understand if they reversed to active cycling after *egr*-expression had ceased. Previous lineage tracing of *rnGAL4* positive cells surviving *egr*-expression indicated that mitotic rates increase 24 hr into the recovery period, which is followed by an increase in total volume of this population by 48 hr (please refer to Figure 1A,D,F in La Fortezza et al., 2016). To independently confirm that G2-arrested cells resume mitotic activity and cellular growth, we analyzed how JNK and cell cycle activity changed during the recovery period. When we analyzed *TRE*-activity in *egr*-expressing discs at 24 hr into the recovery period, we observed decreasing but still pronounced JNK activity (*Figure 2G-H'*). FUCCI analysis and EdU incorporation indicated the presence of G2 cells but also of isolated events of DNA replication activity at the center of the pouch (*Figure 2J,J'*). However, 72 hr into the recovery period, *TRE*-reporter activity was strongly reduced (*Figure 2I,I'*), and FUCCI as well as EdU incorporation assays revealed actively cycling cells (*Figure 2K,K'*). Combined, these results suggest that JNK-induced stalling of the cell cycle in G2 is reversible. However, unlike surgical injury where JNK-activity declined by 16 hr, stalling persisted much longer in *egr*-expressing discs. This temporal correlation between JNK activity and G2-stalling is also reflected by dose-sensitive responses. Specifically, *TRE*-reporter activity scaled with the proportion of cells in G2 and also with stalling-associated increase in cell size (*Figure 2L,L'*). We thus suggest that injury-induced stalling of cells in G2 represent a dose-dependent response to spatio-temporal JNK activity. For our study, we will use the term *stalling* to refer to a transient G2 shift induced by temporally limited JNK activation and use the term *arrest* when we want to emphasize prolonged stalling of G2 in response to high and persistent JNK activity.

## JNK activity is necessary and sufficient for G2 stalling

The strong correlation between JNK-activity and a G2-dominated profile indicated a direct regulation of G2-stalling by JNK. To test this hypothesis, we transiently expressed a constitutively active form of JNKK Hemipterous (*hep^{ACT}*) in the disc pouch using *rnGAL4*. FUCCI analysis indeed revealed a cell cycle shift towards G2 and absence of DNA replication (*Figure 3A,A'*), indicating that JNK is sufficient to induce G2-stalling. Testing the necessity of JNK for G2-stalling in *egr*-expressing disc is challenging, because interference with JNK abolishes the apoptotic stimuli mediating *egr*-induced cell death (*La Fortezza et al., 2016*). To circumvent this issue, we applied acute surgical injury to wing imaginal discs expressing a dominant-negative form of the JNK Basket (*bsk^{DN}*) in the posterior compartment using *enGAL4*. As expected, inhibition of JNK blocked upregulation of the JNK-reporter *puc-LacZ* in the posterior compartment of injured discs (*Figure 3B*). Importantly, it also prevented a cell cycle shift towards G2 at the site of injury (*Figure 3B'*). Taken together, these data demonstrate that JNK-signaling in response to tissue damage is sufficient and necessary to induce G2-stalling.

To test if G2 stalling was a general response to JNK activation, we investigated if developmentally patterned JNK activity was associated with cell cycle changes. JNK in the wing peripodium is

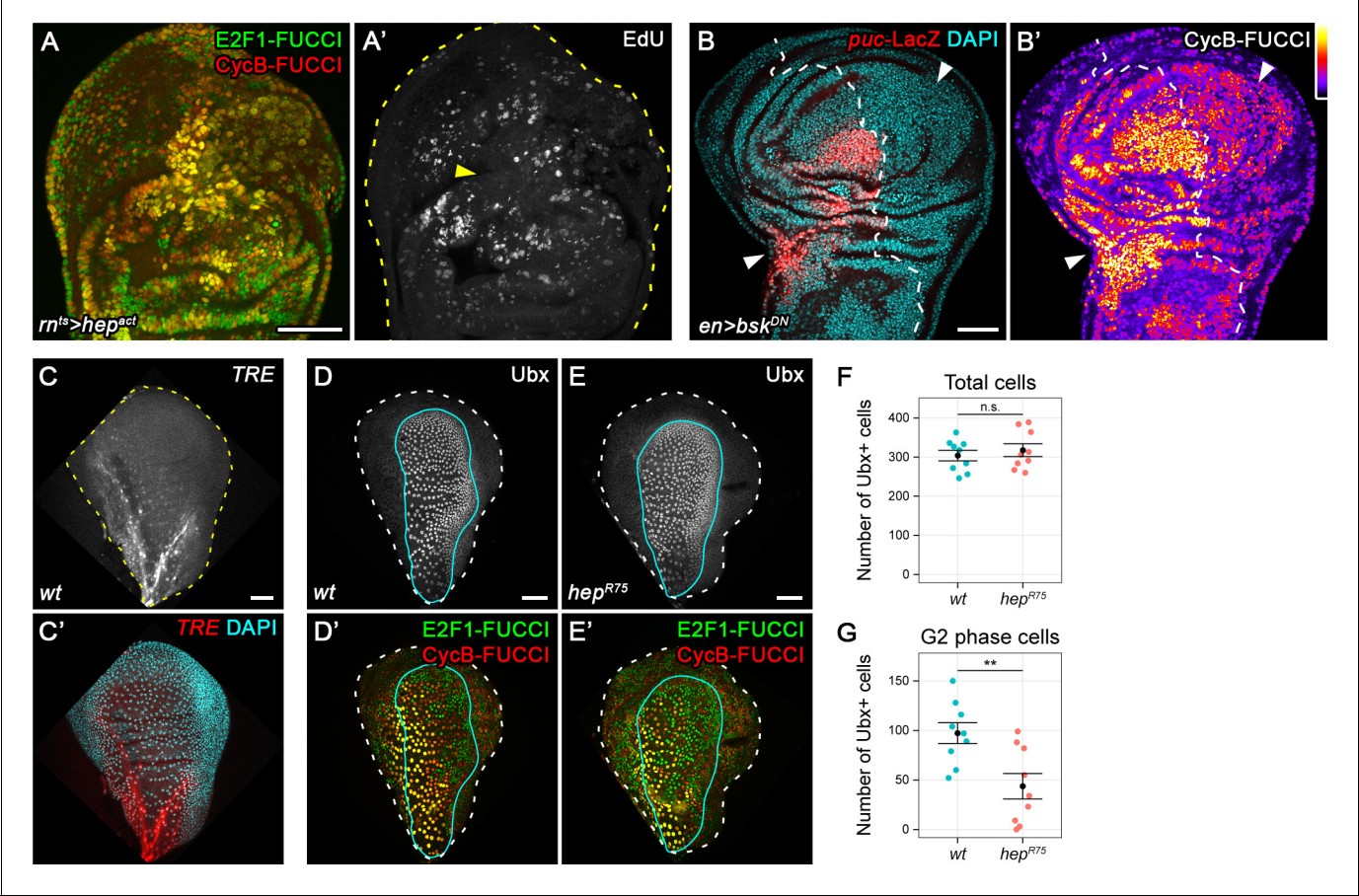

**Figure 3.** JNK activity is necessary and sufficient for G2-stalling. (**A,A'**) A wing disc expressing a constitutively active JNKK *hep^ACT* in the pouch, assayed by FUCCI reporters (**A**) and EdU incorporation (**A'**) for cycling cells. At the center of the pouch, a G2-shifted cell population lacks EdU incorporation (arrow in **A'**). (**B,B'**) A wing disc with surgical damage 6 hr into the recovery period and expressing *bsk^DN* in the posterior compartment (on the right-hand side of the dotted line) under control of *engrailed(en)GAL4*. Wing discs were counterstained with DAPI (cyan) and express the JNK-reporter *puc-LacZ* (red) as well as the G2-specific FUCCI reporter *mRFP-NLS-CycB^1-266* (thermal LUT). Arrows indicate axis of surgical injury verified by tissue deformation in basal sections. (**C,C'**) The periodium of a wild type disc counterstained with DAPI (cyan in **C'**) and expressing the JNK-reporter *TRE*-RFP (**C**, red in **C'**). JNK signaling in the wing periodium is required for wing eversion at the larval-pupal transition (*Pastor-Pareja et al., 2004*). (**D–E'**) The periodium of size-matched wild type (**D,D'**) and *hep^R75* hemizygous mutant discs (**E,E'**) stained for Ubx (gray in **D,E**, outlined in cyan) and expressing both FUCCI reporters (**D',E'**). (**F–G**) Quantifications of the total number of Ubx-positive cells (**F**) and of the Ubx-positive cells in G2 (**G**) in wild type and *hep^R75* hemizygous mutant discs. Graphs display mean ± SEM for *wt*, n = 9 and *hep^R75*, n = 9 discs. *U*-tests were performed to test for statistical significance, n.s. = non significant, \*\*p=0.011. Maximum projections of multiple periodial confocal sections are shown in **C-E'**. Scale bars: 50 µm.

DOI: https://doi.org/10.7554/eLife.41036.006

The following figure supplements are available for figure 3:

**Figure supplement 1.** JNK activity is necessary and sufficient for G2-stalling.
DOI: https://doi.org/10.7554/eLife.41036.007

**Figure supplement 2.** DNA damage is not rate-limiting for G2 stalling.
DOI: https://doi.org/10.7554/eLife.41036.008

required for disc eversion (*Pastor-Pareja et al., 2004*). Strikingly, *TRE* activity in the periodium correlated with a G2 profile and absence of DNA replication (*Figure 3C, C'*, *Figure 3—figure supplement 1A,A'*). To test if JNK was necessary for this developmentally regulated G2 profile, we suppressed JNKK activity in the entire larva by hemizygosity for *hep^R75*, a pupal lethal allele of *hep* (*Glise et al., 1995*). Indeed, the periodium of size-matched *hep^R75* discs displayed a significant reduction in the number of G2 cells and an increase in G1-phase cells, if compared to wild type discs (*Figure 3D-G*, *Figure 3—figure supplement 1B-E*). Combined, these observations indicate that JNK

is at least partially necessary for one example of developmentally regulated G2-stalling in the absence of tissue damage.

The observation that JNKK was required for developmentally patterned G2 stalling suggested that JNK itself rather than tissue damage per se induces these cell cycle changes. In agreement with this hypothesis, we observed no correlation between the occurrence of γH2Av, a marker of dsDNA breaks (*Khurana and Oberdoerffer, 2015*), and G2-stalling in *egr*-expressing discs (*Figure 3—figure supplement 2A-A''*). Moreover, neither knock-down of the DNA damage sensing and relay components *chk1 (grp)*, ATR (*mei-41*), nor organismal hemizygosity for *mei-41$^{RT1}$* prevented the appearance of a G2-dominated FUCCI profile in *egr*-expressing discs (*Figure 3—figure supplement 2B-D*). We conclude that DNA damage per se, normally a potent inducer of G2/M arrests (*Sancar et al., 2004*; *Song, 2005*), is not a rate-limiting driver of G2-stalling. Yet, DNA damage or damage of other cellular components could contribute to G2-stalling via activation of JNK. Importantly, however, activation of JNK could integrate both damage and developmental signals with cell cycle control.

## Regulation of Cdc25/String and Tribbles is rate-limiting for G2 stalling

We next aimed to understand which cell cycle regulator may be targeted by JNK to promote G2-stalling. Knock-down of *Cdk1* in the wing pouch induced a dramatic shift of the FUCCI profile, resembling the shift observed in *egr*-expressing discs (*Figure 4—figure supplement 1A,B*). This suggests that a lack of Cdk1 activity may arrest JNK-signaling cells. A rate limiting activator of Cdk1 during the G2/M transition is the Cdc25-type phosphatase String (Stg) (*Edgar et al., 2001*; *Kimelman, 2014*; *Kiyokawa and Ray, 2008*), whose proteasomal degradation is regulated by Tribbles (Trbl) (*Mata et al., 2000*; *Seher and Leptin, 2000*). We first analyzed a GFP trap inserted in the *stg* locus (*Buszczak et al., 2007*). Of note, the *stg*-GFP chromosome is lethal and GFP expression fails to track with cell cycle phase in individual cells (*Figure 4—figure supplement 1E-E''*), suggesting that the trap disrupts *stg* function and does not give rise to a Stg-GFP fusion protein that is faithfully degraded during the cell cycle. However, expression of GFP from the locus reflects previously published tissue-level expression patterns of *stg* transcripts (*Figure 4—figure supplement 1F-G'''*) (*Johnston and Edgar, 1998*; *Thomas et al., 1994*), indicating that the trap behaves as a reporter of *stg* transcription. We observed that expression of the *stg*-GFP trap was dramatically downregulated in G2-shifted cells of *egr*-expressing discs (*Figure 4A-B'*) suggesting that *stg* transcription was reduced. A GFP-tagged Tribbles protein (*Nagarkar-Jaiswal et al., 2015*; *Otsuki and Brand, 2018*) was highly upregulated in *egr*-expressing and surgically injured discs (*Figure 4C-E*) in a manner that was dependent on JNK activity (*Figure 4E*). Thus, JNK-activity seperately impinges on *stg* transcription and Trbl availability, potentially underlying stalling of cells in G2 in parallel pathways.

We thus wanted to test, if targeted overexpression of *stg* or knockdown of *trbl* can suppress JNK-induced G2-stalling. While *stg* overexpression and *trbl* RNAi-mediated knockdown in the wild type disc reduces developmental G2-patterns at the dorsal-ventral compartment boundary (*Figure 4—figure supplement 1C,D*), we and previous studies (*Mata et al., 2000*; *Neufeld et al., 1998*; *Reis and Edgar, 2004*) failed to detect pronounced alterations in proliferation patterns (*Figure 4—figure supplement 1H-I''*) or adult wings (*Figure 4—figure supplement 1J-L*), suggesting that overexpression of *stg* or knockdown of *trbl* causes little changes to wing development. Importantly, overexpression of *stg* and or knockdown of *trbl* in *egr*-expressing cells re-established a heterogeneous FUCCI cell cycle profile (*Figure 4F-H*). Specifically, cycling cells in *egr,stg*-coexpressing domains could be detected by EdU incorporation and pH3 labeling (*Figure 4I-J'*). Combined, these observations suggest that *trbl* and *stg* are rate limiting for cell cycle progression in JNK-signaling cells and that overexpression of *stg* or knockdown of *trbl* is sufficient to override damage-induced cell cycle stalling in G2.

## Chronic stalling in G2 interferes with proliferative capacity

Having identified two potent suppressors of G2-stalling, we asked what role stalling may have in tissue stress responses. We focus in the following experiments on *stg*, as a more direct cell cycle effector. *egr*-expressing discs normally present with a folded architecture (*Figure 5A,B*) and large G2-arrested cells at the center of the pouch (*Figure 2A,B*). In contrast, *egr,stg*-co-expressing discs presented with densely and regularly arranged columnar cells (*Figure 5A-C'*). Moreover, *stg* co-

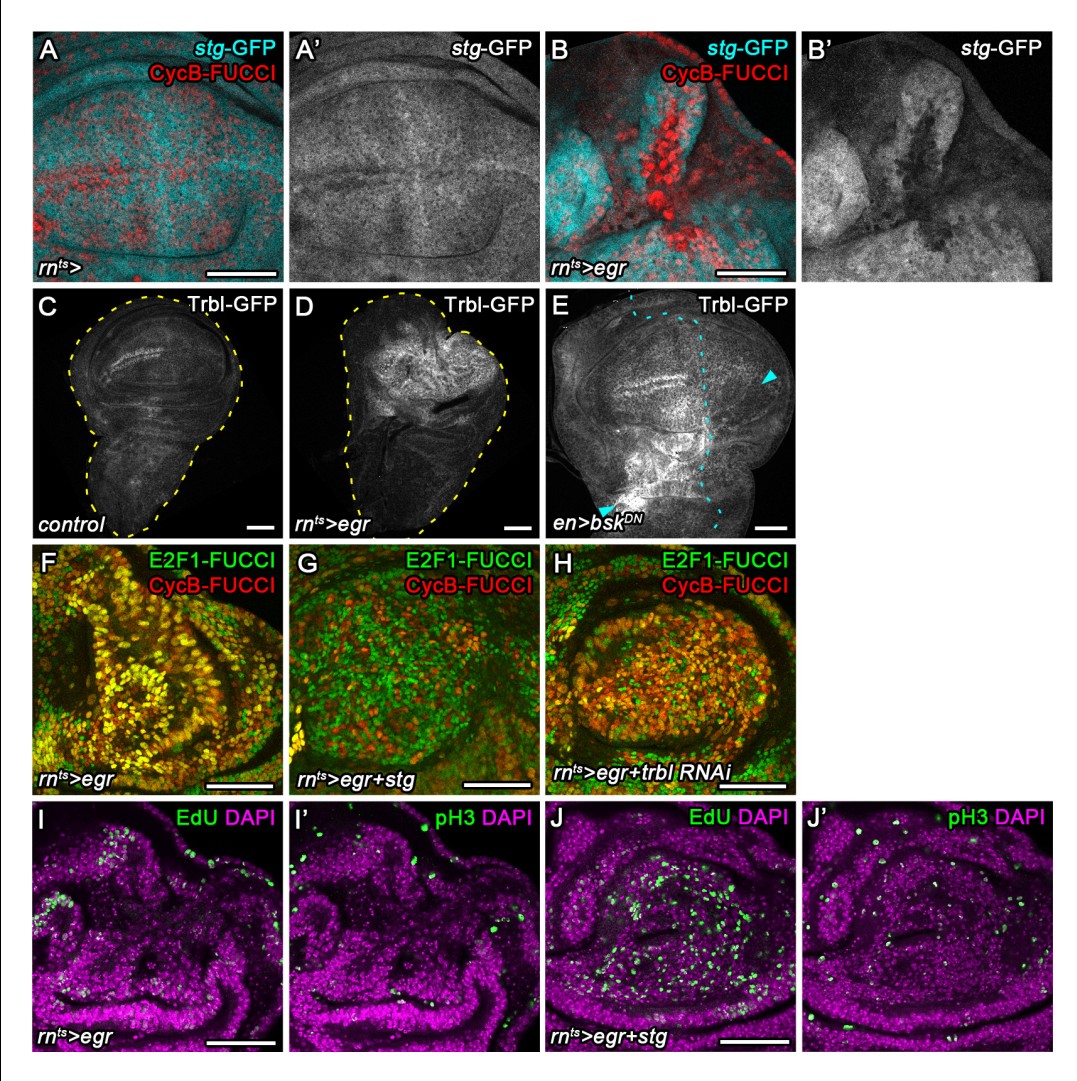

**Figure 4.** Cdc25/String and Tribbles regulate G2 stalling. (**A–B'**) Control wing disc (**A,A'**) and *egr*-expressing disc at 0 hr into the recovery period (**B,B'**) expressing a GFP trap element in the *stg* locus (**A',B'**, cyan in **A,B**) and the G2-specific FUCCI reporter *mRFP-NLS-CycB$^{1-266}$* (red in **A,B**). Note pronounced reduction of GFP expression in G2-arrested cells at the center of the pouch (**B,B'**). (**C–E**) Control wing disc (**C**), an *egr*-expressing disc at 0 hr into the recovery period (**D**), and a surgically damaged wing disc 6 hr into the recovery period expressing *bsk$^{DN}$* in the posterior compartment (on the right-hand side of the dotted line) under control of *enGAL4* (**E**). All discs also express a GFP-tagged Trbl protein expressed from the native locus (***Nagarkar-Jaiswal et al., 2015***). Arrows indicate axis of surgical injury verified by tissue deformation in basal sections. (**F–H**) An *egr*-expressing disc (**F**), an *egr,stg*-co-expressing disc (**G**) and an *egr,trbl RNAi*-co-expressing disc at 0 hr into the recovery period expressing the FUCCI reporters. Note increase in the frequency of G1 cells in (**G,H**). (**I–J'**) An *egr*-expressing disc (**I,I'**) and an *egr,stg*-co-expressing disc (**J,J'**) analyzed by EdU incorporation to reveal DNA replication activity (green in **I,J**) and by staining for phospho-Histone3 to reveal mitotic cells (pH3) (green in **I',J'**). Discs were counterstained with DAPI (magenta). Note increase in the frequency of S- and M-phase cells upon *egr,stg* co-expression. Maximum projections of multiple confocal sections are shown in **F-H**. Scale bars: 50 µm.

DOI: https://doi.org/10.7554/eLife.41036.009

The following figure supplement is available for figure 4:

**Figure supplement 1.** Cdc25/String and Tribbles regulate G2 stalling.
DOI: https://doi.org/10.7554/eLife.41036.010

expression improved tissue regeneration in *egr*-expressing discs, as assessed by size and

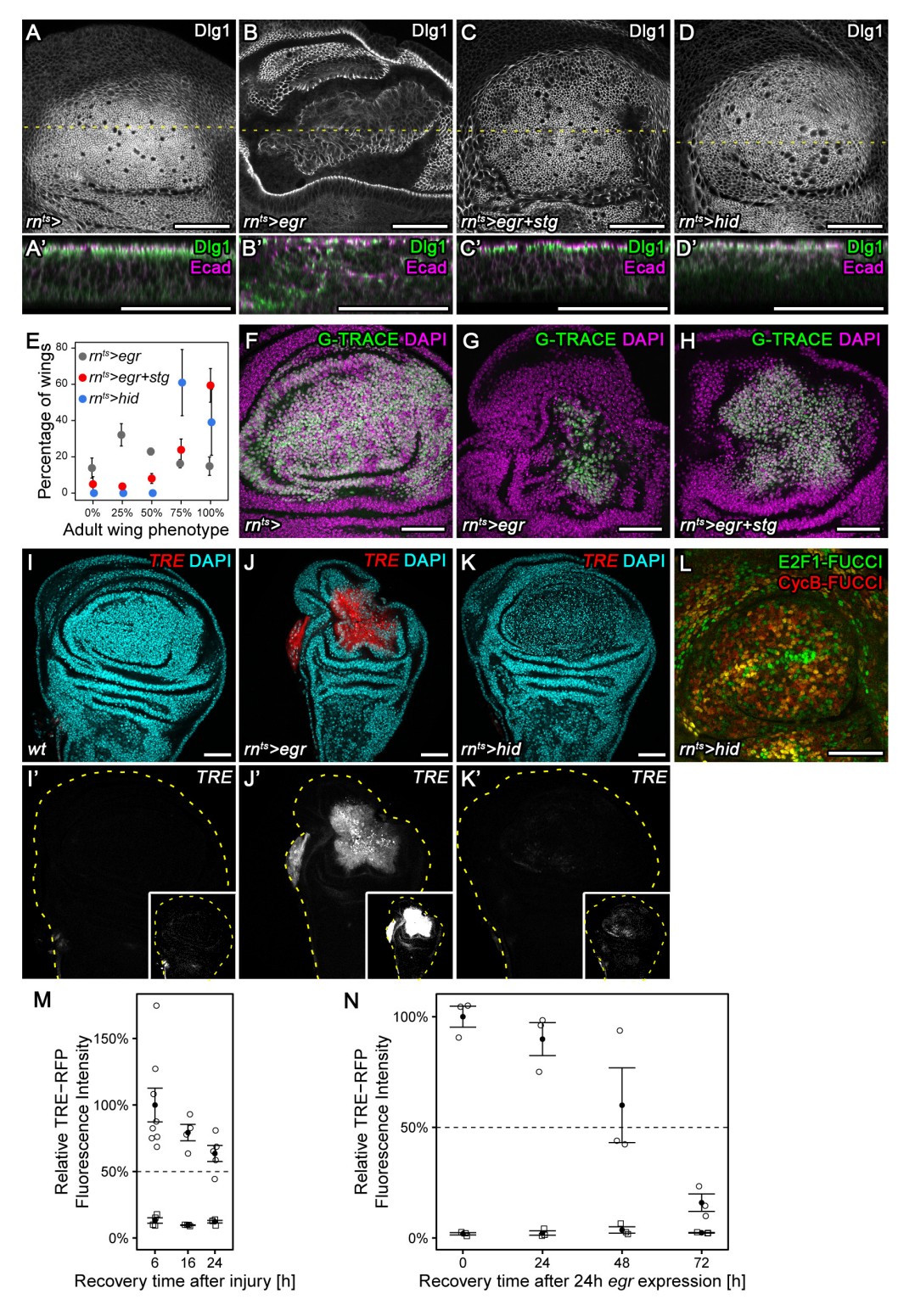

**Figure 5.** Chronic stalling in G2 interferes with proliferative capacity. (A–D') X-Y view of a control wing disc (A), an *egr*-expressing (B), *egr,stg*-co-expressing (C) or *hid*-expressing disc (D) at 0 hr into the recovery period. Cross-sections through the tissue (A'–D') were visualized along dotted yellow lines. Discs were stained for Discs large 1 (Dlg1, A-D, green in A'-D') and E-cadherin (Ecad, magenta in A'-D') to visualize cell outlines and cell polarity. (E) Adult wings developing from *egr*-expressing, *egr,stg*-co-expressing or *hid*-expressing discs were classified
*Figure 5 continued on next page*

*Figure 5 continued*

according to wing size and morphology (see Materials and methods, *Figure 5—figure supplement 1A-C*). Graphs display mean ± SEM of ≥3 independent experiments. Note the significantly improved wing regeneration of *rn^ts^>egr+stg* (p<0.0001, n = 676 wings) and *rn^ts^>hid* (p<0.0001, n = 514 wings) when compared to *rn^ts^>egr* (n = 718 wings) by chi-squared tests. (F–H) Control wing disc (F), an *egr*-expressing (G) and *egr,stg*-co-expressing (H) disc at 0 hr into the recovery period where the surviving *rnGAL4*-lineage has been labeled by G-TRACE (green) (*Evans et al., 2009*). Discs were counterstained with DAPI (magenta). (I–K') Control wing disc (I), an *egr*-expressing (J) or *hid*-expressing (K) disc at 0 hr into the recovery period. Discs express the JNK-reporter *TRE*-RFP (I'-K', red in I-K) and were counterstained with DAPI (cyan in I-K). *TRE*-reporter activity was imaged at settings optimized to subsaturation in *egr*-expressing discs. Small insets in (I'–K') show the same images adjusted to the dynamic range in *hid*-expressing discs. Note that distinct DAPI dense particles seen in the pouch of *hid*-expressing discs represent remnants of apoptotic cells. (L) A *hid*-expressing disc at 0 hr into the recovery period expressing FUCCI reporters (compare to *Figure 2B*). (M,N) Quantifications of *TRE*-RFP fluorescence intensity at the wound site in surgically injured wing discs at 6 hr, 16 hr, and 24 hr after tissue damage (M, circles) and in *egr*-expressing discs at 0 hr, 24 hr, 48 hr and 72 hr into the recovery period (N, circles). Larvae with surgically injured wing discs pupariate at 24 hr so later time points could not be quantified. Note that TRE-RFP reporter activity declines faster in surgically injured discs. Fluorescence intensity in non-wound regions (squares) serves as baseline reference. Graphs display mean ± SEM for n = 8 (6 h), n = 4 (16 h), n = 5 (24 h) injured discs (M) or n = 3 (0 h), n = 3 (24 h), n = 3 (48 h), n = 3 (72 h) *egr*-expressing discs (N). Maximum projections of multiple confocal sections are shown in A-D, F-H. Scale bars: 50 µm.

DOI: https://doi.org/10.7554/eLife.41036.011

The following figure supplement is available for figure 5:

**Figure supplement 1.** Chronic stalling in G2 interferes with proliferative capacity.
DOI: https://doi.org/10.7554/eLife.41036.012

morphology of adult wings. 59% of adult wings developing from *egr,stg*-co-expressing discs were of wild type size, in contrast to just 14% of wings developing from *egr*-expressing discs (*Figure 5E*, *Figure 5—figure supplement 1A,B*). Importantly, *stg* expression did not interfere with activation of apoptosis (*Figure 5—figure supplement 1D-G*) or JNK activation (*Figure 5—figure supplement 1H*) in response to *egr* expression. Instead, the *egr,stg*-expressing cell population labelled by G-TRACE (*Evans et al., 2009*) was larger in size (*Figure 5F-H*). This suggests that *stg* overexpression did not interfere with cell ablation but specifically with arrest of the cell cycle and thus proliferative capacity in *egr*-expressing cells. *egr*-expressing discs had been previously reported to exhibit low regenerative potential, in contrast to discs expressing the pro-apoptotic gene *hid* (*Herrera et al., 2013*), an antagonist of *Diap1*-dependent inhibition of caspase activity (*Vaux and Silke, 2005*). Strikingly, preventing a G2-arrest in *egr*-expressing discs by *stg* overexpression phenocopied *hid*-induced regeneration, where discs display columnar morphology (*Figure 5D,D'*) and regenerate efficiently to normal-sized adult wings (*Figure 5E*, *Figure 5—figure supplement 1C*). Importantly, *hid*-expressing discs activated the *TRE*-reporter only at low levels and forwent the pronounced changes to FUCCI reporter activity observed in *egr*-expressing discs (*Figure 5I-L*). These observations indicate that persistent stalling and arrest of cells in G2 interferes with regeneration in *egr*-expressing discs by interfering with the ability of cells to divide and proliferate. Moreover, they highlight G2-stalling as dose-sensitive JNK-effector which determines the regenerative potential of different experimental regeneration models. Of note, *egr*-expression also induced TRE-reporter activity at much higher levels (Figure 7B,B',D,D') and at longer timescales than surgical injury (*Figure 5M,N*), supporting our conclusion that *egr*-expressing discs may experience extreme and possibly aberrant G2 stalling and arrest.

## Stalling in G2 promotes survival by protecting cells from JNK-induced apoptosis

Our observation that G2-stalled cells reenter the cell cycle when JNK-signaling decreases suggests that G2-stalled cells can survive in high, potentially pro-apoptotic JNK-signaling environments. We therefore asked if G2-stalled cells are resistant to apoptosis. We thus suppressed G2-stalling in surgically injured discs by expressing *stg* using *rnGAL4*. Strikingly, 6 hr after injury, we observed a 2-fold increase in the apoptotic domain within injured discs (*Figure 6A-E*). Importantly, while *stg* overexpression also induced apoptosis primarily at the anterior D/V boundary in undamaged control discs,

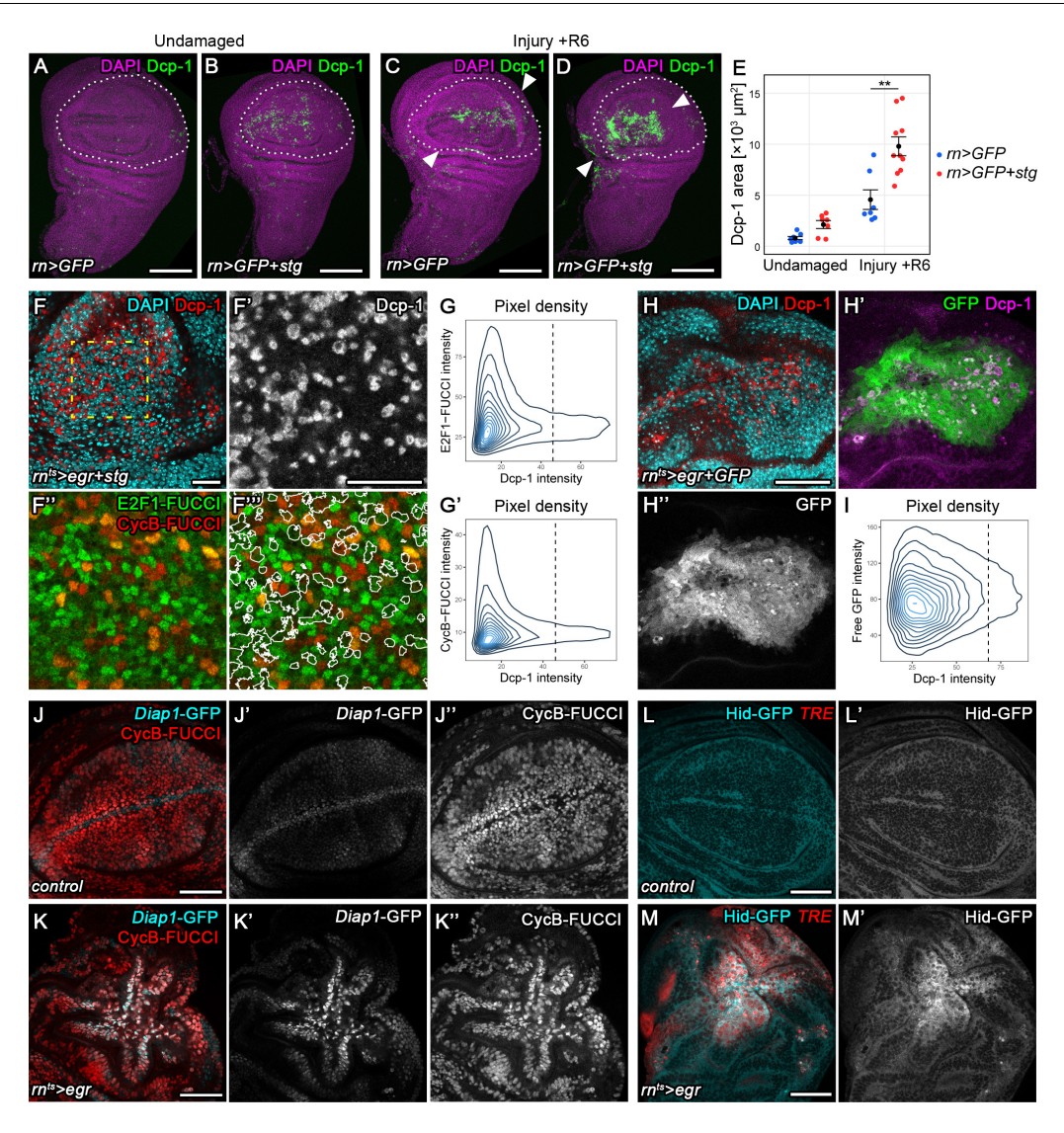

**Figure 6.** Transient stalling in G2 promotes survival by protecting cells from JNK-induced apoptosis. (A–E) Undamaged control (A) and an undamaged disc expressing *stg* under the control of *rnGAL4* (B). An injured control disc (C) and an injured *stg*-expressing disc (D), 6 hr into the recovery period. Dotted lines indicate the *rnGAL4* domain, arrows indicate injury axis. Wing discs were stained for the apoptotic marker Dcp-1 (green) and counterstained with DAPI (magenta). The area occupied by Dcp-1 in a maximum projection is quantified in (E). Graphs display mean ± SEM for undamaged *rn>GFP*, n = 8; undamaged *rn>GFP+stg*, n = 7; injured *rn>GFP*, n = 7; injured *rn>GFP+stg*, n = 10 discs. *U*-tests were performed to test for statistical significance, \*\*p<0.01. (F–F''') An *egr,stg*-expressing disc analyzed for FUCCI activity (F'',F''') and the apoptotic marker Dcp-1 (F', red in F) or DAPI (cyan in F) at 0 hr into the recovery period. F'-F''' represent the area framed by broken line in F. White lines in F''' represent the Dcp-1 outline mask of F'. Note that masked cells generally express low levels of either FUCCI reporter. (G–G') The fluorescence intensities of Dcp-1 and the *GFP-E2f1$^{1-230}$* (G) or *mRFP-NLS-CycB$^{1-266}$* (G') FUCCI reporters for each pixel are plotted as 2D density graphs (see Materials and methods). The broken line represents a visually chosen Dcp-1 threshold defining apoptotic cells. Note that the surviving population expresses the entire range of FUCCI reporter intensities (left), in contrast to apoptotic cells (right). (H–I) An *egr,GFP*-expressing disc stained for the apoptotic marker Dcp-1 (red in H, magenta in H') and DAPI (cyan in H) at 0 hr into the recovery period. Graph (I) plots the 2D density of pixel fluorescence intensities for Dcp-1 and free GFP. The broken line represents a visually chosen Dcp-1 threshold defining apoptotic cells. Note that the surviving and apoptotic population use the GFP fluorescence spectrum symmetrically. (J–K'') Control (J–J'') and an *egr*-expressing (K–K'') disc at 0 hr into the recovery period expressing the *Diap1-GFP.3.5* reporter (J',K', cyan in J,K) and the G2-specific FUCCI reporter *mRFP-NLS-CycB$^{1-266}$* (J'',K'', red in J,K). Note the anti-correlation between *Diap1* promoter activity and G2-phase in control discs, in contrast to *egr*-expressing discs. (L–M') Control (L,L') and *egr*-expressing (M,M') discs at 0 hr into the recovery period expressing a Hid-GFP fusion protein under endogenous control (L',M', cyan in L,M). JNK-signaling cells were detected by activation of *TRE*-RFP (red in L,M). Maximum projections of multiple confocal sections are shown in A-D. Scale bars: 100 μm (A–D), 20 μm (F,F'''), 50 μm (H–M).

DOI: https://doi.org/10.7554/eLife.41036.013

*Figure 6 continued on next page*

*Figure 6 continued*

The following figure supplement is available for figure 6:

**Figure supplement 1.** Stalling in G2 promotes survival by protecting cells from JNK-induced apoptosis.

DOI: https://doi.org/10.7554/eLife.41036.014

apoptosis in surgically injured discs was specifically increased near the wound site where JNK reporters are expected to be activated (*Figure 6B,D*). This suggests that at transient time scales, G2-stalling is required to prevent apoptosis and thus promotes survival in injured tissues displaying potentially lethal levels of JNK activity.

If stalling in G2 protects cells from apoptosis, are JNK-signaling cells more susceptible to apoptosis in other phases of the cell cycle? To test this idea, we first confirmed that actively cycling, JNK-signaling cells did not die in late G2 by demonstrating that Dcp-1 levels were highest in *egr*-expressing cells with low levels of a co-expressed HA-tagged *stg* (*Figure 6—figure supplement 1A-A'''*). Stg-HA peaked in late G2 and mitotic cells but was absent in G1, S-phase and early G2 (*Figure 6—figure supplement 1B-B''*). To further narrow down the cell cycle phase at which JNK signaling cells died, we correlated levels of the apoptotic marker Dcp-1 with FUCCI reporter activity in *egr,stg*-co-expressing discs. We took care to specifically analyze confocal sections within the tissue to catch apoptotic cells before extrusion. We found that in actively cycling, JNK-signaling tissue, Dcp-1 could be specifically detected in cells with low intensity of either FUCCI reporter (*Figure 6F-G'*), suggesting that JNK-signaling cells preferentially died either early in G1 or late in S-phase. Importantly, low FUCCI reporter fluorescence was not due to apoptosis-dependent degradation of fluorophores, as levels of fluorescence intensity of an unrelated cytoplasmic GFP were independent of Dcp-1 levels in *egr,GFP*-co-expressing discs (*Figure 6H-I*). Additionally, the overall FUCCI profile of *egr,stg* and *stg*-expressing control discs was similar, confirming that FUCCI reporter intensities were not affected by apoptotic cells per se (*Figure 6—figure supplement 1C-C''*). To better understand when cycling JNK-signaling cells died, we allowed discs to incorporate EdU for at least 1 hr prior to fixation. We consistently failed to detect recent DNA replication in apoptotic cells (*Figure 6—figure supplement 1E-E''*) indicating that cells did not die in S-phase. Taken together, these results support a model where JNK-signaling cells are susceptible to apoptosis in G1 and protected from apoptosis by stalling in G2.

To understand how G2-stalled cells may be protected from apoptosis, we analyzed expression of *Diap1*, an inhibitor of the initiator caspases, as well as of *hid*, a potent IAP antagonist (*Vasudevan and Ryoo, 2015*; *Vaux and Silke, 2005*). A *Diap1*-GFP reporter encompassing enhancer elements sensitive to regulation by anti-apoptotic Hippo/Yorkie signaling (*Zhang et al., 2008*) was upregulated in *TRE*-positive (data not shown) and G2-stalled cells at the center of *egr*-expressing discs (*Figure 6J-K''*). Importantly, *Diap1*-GFP activity did not correlate with tissue patterns of G2 in undamaged control discs (*Figure 6J,J''*), suggesting that Yorkie activity at *Diap1* regulatory elements was not controlled in a cell cycle-dependent manner per se, but likely reflected activation of Yorkie by JNK (*Sun and Irvine, 2011*; *Ríos-Barrera and Riesgo-Escovar, 2013*). A Hid-GFP fusion protein expressed under the control of endogenous regulatory elements (*Nagarkar-Jaiswal et al., 2015*) was also strongly upregulated in *TRE*-positive cells in *egr*-expressing discs (*Figure 6L-M'*). These observations align with previous reports of activation of anti-apoptotic Yorkie (*Sun and Irvine, 2011*; *Ríos-Barrera and Riesgo-Escovar, 2013*) or pro-apoptotic Hid (*Luo et al., 2007*; *Shlevkov and Morata, 2012*) by JNK in stressed cells. We highlight that the activation of anti-apoptotic and pro-apoptotic pathways occurs concomitantly in JNK-signaling G2-stalled cells. Taken together, these results suggest that transition from G2 to G1 represents the key switch in the cellular interpretation of opposing JNK-dependent signals with anti-apoptotic to pro-apoptotic consequences.

## Chronic stalling in G2 promotes non-autonomous overgrowth

Cell cycle arrest and apoptosis resistance are hallmarks of senescence. Senescent cells affect their microenvironment through senescence-associated secretory phenotype (SASP) (*Hernandez-Segura et al., 2018*; *Neves et al., 2015*; *Pluquet et al., 2015*; *Salama et al., 2014*), which promotes tumorigenesis and contributes to tumor heterogeneity (*Hinds and Pietruska, 2017*;

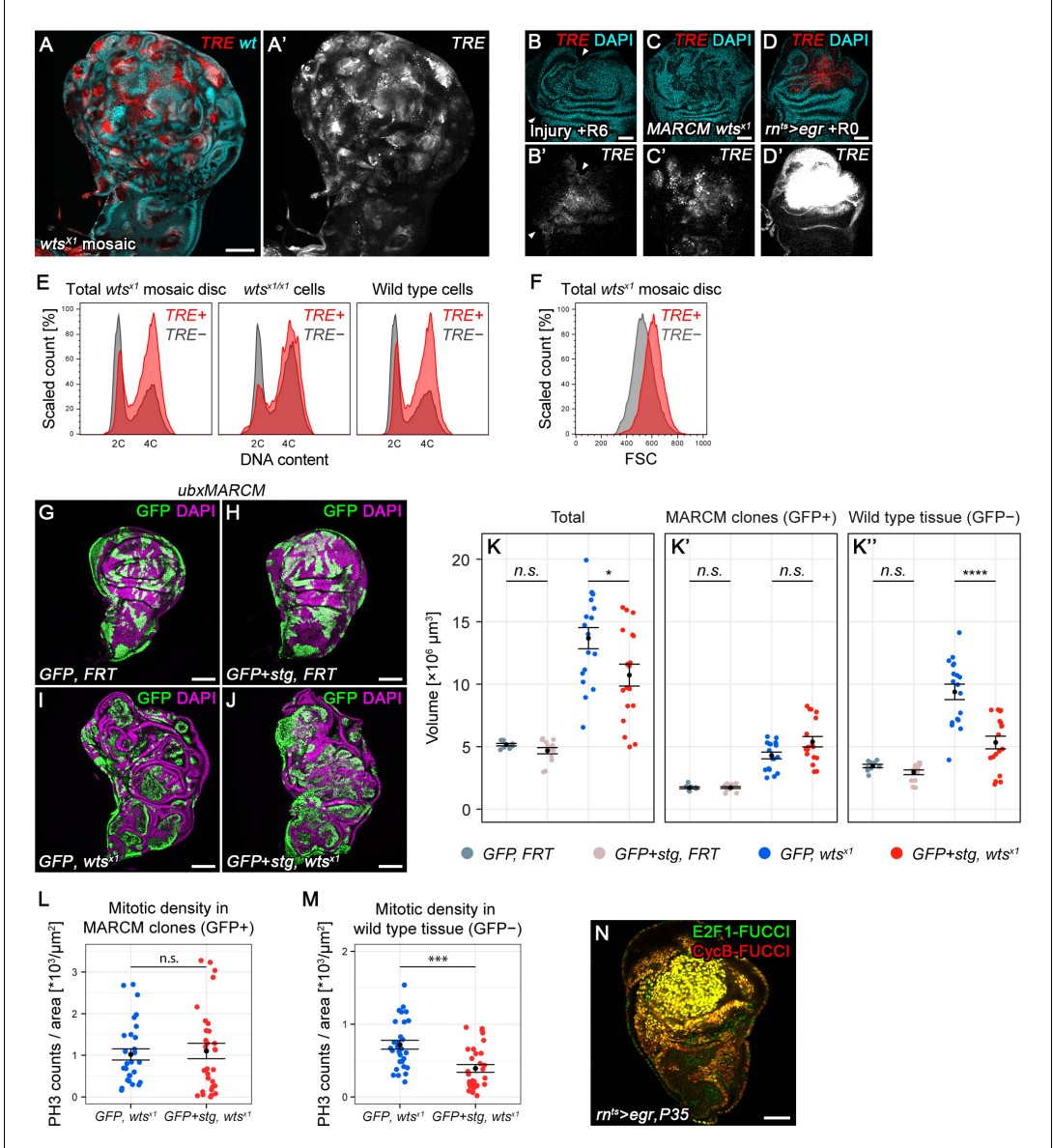

**Figure 7.** Chronic stalling in G2 promotes non-autonomous overgrowth. (A,A') A wing disc expressing *TRE*-RFP and carrying mosaic *wts*^x1/x1 clones marked by the absence of GFP (cyan in **A**). (**B–D'**) A surgically damaged wing disc 6 hr into the recovery period (**B,B'**), a wing disc carrying *wts*^x1/x1 MARCM clones (**C,C'**) and an *egr*-expressing disc at 0 hr into the recovery period (**D,D'**). Discs express the JNK-reporter *TRE*-RFP (**B'-D'**, red in **B-D**) and were counterstained with DAPI (cyan in **B-D**). *TRE*-reporter activity was imaged at settings optimized to subsaturation in *egr*-expressing discs. Panels (**B'–D'**) show the *TRE*-RFP fluorescence adjusted to the dynamic range of surgically injured discs. (**E–F**) Mosaic *wts*^x1/x1 wing discs were analyzed for DNA content (**E**) and cell size (**F**) by flow cytometry. The total cell population of discs was plotted as *TRE*-positive or *TRE*-negative events (**E,F**). The same analysis was also applied separately to *wts*^x1/x1 cells and wild type cells sub-populations (**E** only). Of note, previous cell cycle studies of Hippo-pathway mutant have not reported any alterations (**Harvey et al., 2003**; **Huang et al., 2005**; **Tapon et al., 2002**). Therefore, the mild cell cycle shift in TRE-negative *wts*^x1/x1 cells appears to be specific to the *wts*^x1 allele. (**G–J**) Wing imaginal discs carrying GFP-labeled MARCM clones (green) that are either wild type (**G**), *stg*-overexpressing (**H**), mutant for *wts*^x1 (**I**) or *stg*-overexpressing and mutant for *wts*^x1 (**J**). Discs were counterstained with DAPI (magenta). (**K–K''**) Volumes occupied by the total disc (**K**), the GFP-labeled fraction representing MARCM clones (**K'**) and the non-GFP-labeled fraction representing the surrounding wild type tissue (**K''**). Graphs display mean ± SEM for *tub>GFP, FRT*, n = 9; *tub>GFP+stg, FRT*, n = 13; *tub>GFP, FRT wts*^x1, n = 18; *tub>GFP+stg, FRT wts*^x1, n = 18 discs. *U*-tests were performed to test for statistical significance (*n.s.* not significant, *p<0.05, ****p<0.0001). (**L,M**) Quantification of phospho-Histone3 events identifying mitotic cells, normalized to the relevant tissue area. Mitotic cells were counted in GFP-positive MARCM clones that are either mutant for *wts*^x1 (blue) or mutant for *wts*^x1 and overexpressing *stg* (red) (**L**), and in the non-GFP-labeled fraction representing the wild type tissue surrounding clones mutant for *wts*^x1 (blue) or mutant for *wts*^x1 and overexpressing *stg* (red) (**M**). Graphs display mean ± SEM, n = 30 confocal sections from six discs per sample. *U*-tests were performed to test for statistical significance (*n.s.* not

*Figure 7 continued*

significant, ***p<0.001). (**N**) An *egr,p35*-co-expressing disc at 0 hr into the recovery period expressing both FUCCI reporters. Scale bars: 50 µm (**B–D,N**), 100 µm (**A, G–J**).

DOI: https://doi.org/10.7554/eLife.41036.015

The following figure supplements are available for figure 7:

**Figure supplement 1.** Chronic stalling in G2 promotes non-autonomous overgrowth.

DOI: https://doi.org/10.7554/eLife.41036.016

**Figure supplement 2.** Chronic stalling in G2 promotes mitogenic signaling.

DOI: https://doi.org/10.7554/eLife.41036.017

*Schosserer et al., 2017*). Many studies report heterogenous activation of JNK in imaginal discs upon genetic loss of tumor suppressor function (*Richardson and Portela, 2018*). We thus tested if imaginal disc tumor models displayed any evidence of a JNK-induced G2-shift. We first analyzed *warts (wts^{x1}*) mosaic clones (*Xu et al., 1995*) which promote Yorkie activity, in wing imaginal discs also expressing *TRE*-RFP (*Figure 7A*). Localized *TRE*-RFP expression increased in mosaic discs during larval stages (*Figure 7—figure supplement 1A*). Based on flow cytometry analysis, we estimated that 40.6% of *wts^{x1}* cells had at least elevated *TRE*-RFP reporter activity. Consistent with non-autonomous induction of JNK in response to tissue stress (*Bosch et al., 2005*; *Herrera et al., 2013*; *Wu et al., 2010*) 29.9% of wild type cells displayed *TRE*-RFP reporter activity. Importantly, *TRE*-RFP levels were comparable to levels induced by surgical injury but much lower than those induced by *egr*-expression (*Figure 7B-D'*). Flow cytometry analysis revealed that *TRE*-positive cells exhibited a strong G2 profile which correlated with an increase in cell size (*Figure 7E,F*). This effect was observed in *TRE*-positive *wts^{x1}* and wild type cells (*Figure 7E*). This relationship could also be observed in mosaic discs containing clones mutant for *scrib^{dt12}*, a hypomorphic tumor suppressor mutation in the key component of the Scrib/Lgl/Dlg epithelial polarity module (*Figure 7—figure supplement 1B-D*) (*Stephens et al., 2018*; *Zeitler et al., 2004*). Combined, these observations suggest that a pronounced shift towards G2 is associated with JNK activity in response to tissue stress imposed by the presence of abnormal or tumorigenic cells.

To understand if G2-stalling played a role in tumor growth or tumor microenvironment, we overexpressed *stg* in *wts^{x1}* or *dlg1^{G0342}* clones. Like *scrib*, *dlg1* is a tumor suppressor in the Scrib/Lgl/Dlg epithelial polarity module (*Stephens et al., 2018*) and mutant cells activate JNK (*Igaki, 2009*; *Igaki et al., 2009*). Overexpression of *stg* in mosaic *wts^{x1}* or *dlg1^{G0342}* clones did not significantly increase clone size (*Figure 7G-K''*, *Figure 7—figure supplement 1E-I'*). As division rates of cycling cells are not expected to be enhanced by *stg* overexpression alone (*Neufeld et al., 1998*), promoting cycling of a small fraction of arrested tumor cells may not significantly increase tumor mass.

Strikingly, however, when we further analyzed these mosaic discs, we found that *stg* overexpression in *wts^{x1}* and in *dlg1^{G0342}* cells strongly affected the surrounding wild type tissue. The absolute size of the wild type tissue in mosaic *wts^{x1}* or *dlg1^{G0342}* mutant discs is almost double of that in mosaic wild type control discs (*Figure 7K''*, *Figure 7—figure supplement 1I'*), a phenomenon ascribed to non-autonomous overgrowth stimulated by the chronic presence of tumorigenic cells (*Fuchs and Steller, 2015*; *Pastor-Pareja and Xu, 2013*; *Uhlirova et al., 2005*). However, *stg* overexpression strongly reduced the size of the surrounding wild type tissue in mosaic *wts^{x1}* and *dlg1^{G0342}* discs (*Figure 7K''*, *Figure 7—figure supplement 1I'*). Importantly, whereas mitotic activity in *wts^{x1}* clones was unaffected (*Figure 7L*), *stg*-overexpression strongly reduced mitotic activity in the wild type tissue surrounding *wts^{x1}* clones (*Figure 7M*). This strongly suggests that stalling of JNK-signaling cells in G2 directly promotes non-autonomous proliferation and thus causes non-autonomous overgrowth on prolonged timescales during imaginal disc tumor development.

Previous studies describe 'undead' cells as chronic drivers of non-autonomous overgrowth. Experimentally, undead cells are created by expression the anti-apoptotic factor p35 in apoptotic JNK-signaling cells (*Chen, 2012*; *Fuchs and Steller, 2015*; *Martín et al., 2009*; *Pérez-Garijo et al., 2009*; *Shlevkov and Morata, 2012*; *Wells et al., 2006*). We created undead cells by co-expressing p35 in *egr*-expressing disc. *Egr,p35*-coexpressing cells completely arrest in G2 (*Figure 7N*), confirming that G2-stalling is intimately associated with cellular states known to stimulate non-autonomous growth. Genetically, the initiator caspase Dronc is required to stimulate non-autonomous growth from dying and undead cells (*Enomoto et al., 2015*; *Fan and Bergmann, 2008a*; *Fan and Bergmann, 2008b*;

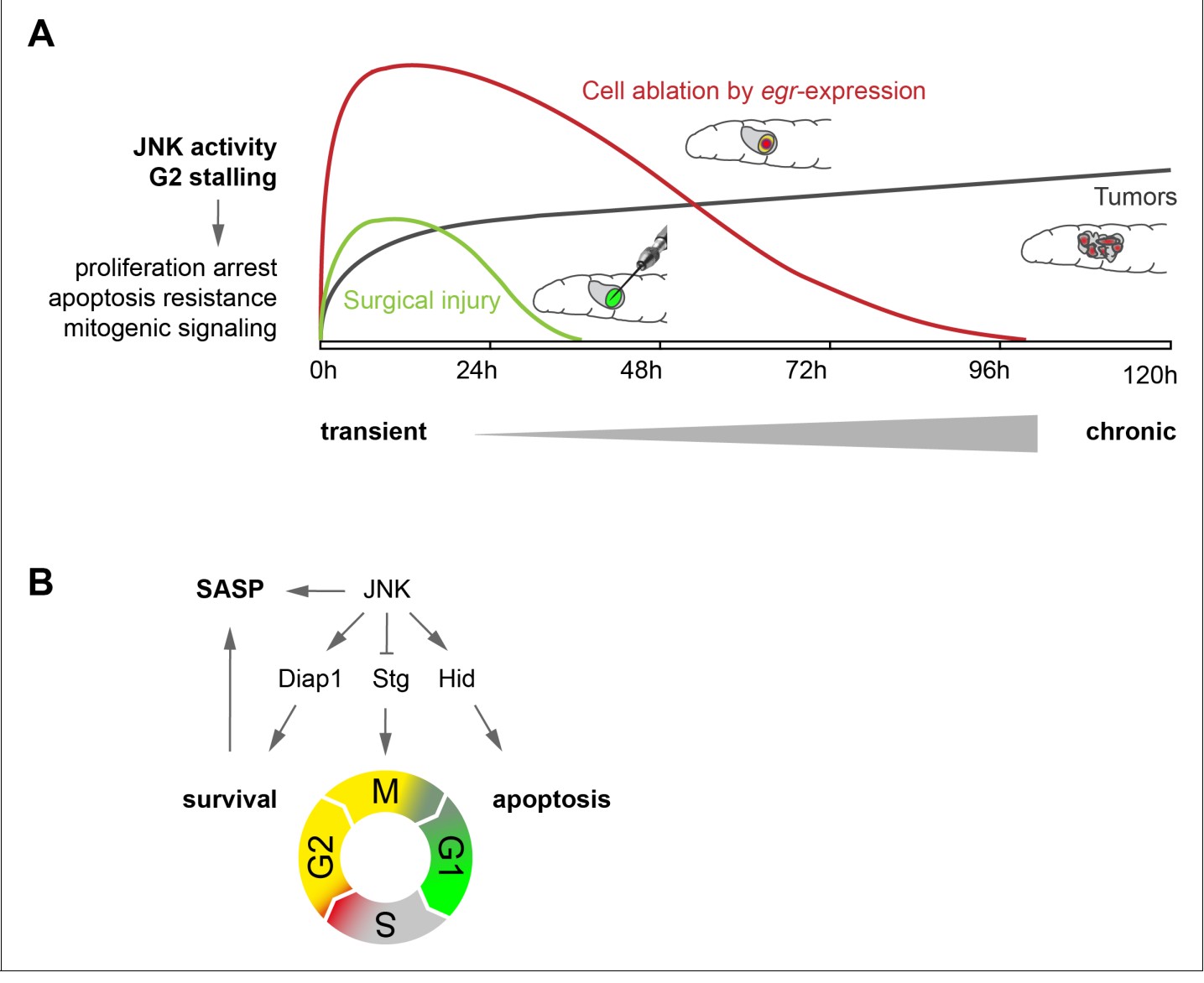

**Figure 8.** Model. (**A**) Transient (surgical injury), prolonged (*egr*-expression) and chronic (mosaic tumors) disruption of tissue homeostasis induces transient, prolonged and chronic JNK activity, thereby driving G2-stalling and senescence-like properties in a dose- and time-dependent manner. (**B**) JNK regulates SASP, Diap1, Stg and Hid. The transition between G2 and G1 acts as switch that prevents survival and SASP. The decision to arrest in G2 is JNK-dependent, which can integrate information about damage and has cell-protective functions. The decision to die in G1 may depend on additional information about the extent of cell and tissue damage.

DOI: https://doi.org/10.7554/eLife.41036.018

*Kiyokawa and Ray, 2008*; *Kondo et al., 2006*; *Wells et al., 2006*). To demonstrate that caspases are activated in G2 stalled cells and that G2 stalling may confer resistance to the execution of apoptosis, we analyzed the CasExpress sensor in surgically injured discs. CasExpress permanently labels anastatic cells, which have survived caspase activation (*Ding et al., 2016*; *Tang et al., 2012*). Strikingly, 24 hr after surgical injury, we observed many CasExpress-positive clones near the wound site (*Figure 7—figure supplement 2A,B*). This indicates that many wound-proximal cells survive caspase activation and subsequently proliferate. While we cannot demonstrate that these anastatic cells corresponded to G2-stalled cells, we suggest that G2 stalling could facilitate transient mitogenic signaling of Dronc-positive, JNK-activated cells at wound sites.

We wanted to understand if G2-arrested cells may affect their microenvironment not just via induction of mitogenic signaling, but also through upregulation of additional SASP-like markers. We

tested reporters for ECM degrading enzymes (MMP1), ROS response (*GstD*-GFP) and UPR (Xbp1-GFP), in addition to reporters for mitogenic signaling (*upd*-LacZ), for activation in *egr*-expressing discs. As expected from many studies previously linking upregulation of these markers to JNK (*Bunker et al., 2015*; *Fulda et al., 2010*; *Richardson and Portela, 2018*; *Santabárbara-Ruiz et al., 2015*; *Takino et al., 2014*; *Uhlirova and Bohmann, 2006*), we found all SASP-like markers, including cell size (*Figure 2—figure supplement 1D*), to be highly elevated (*Figure 7—figure supplement 2C-J*). The co-occurrence of G2-arrest, JNK activity and SASP-like markers suggests that a JNK-signaling induced G2-arrest in flies is linked to senescence-like phenotypes driven by JNK. Importantly, transient *upd*-LacZ and MMP1 upregulation is associated with transient JNK activity in surgical injuries (*Lin et al., 2010*; *McClure et al., 2008*), making it plausible that transient G2-stalling is linked to senescence-like properties promoting wound healing and regeneration, analogous to senescent cells observed in a mammalian wound model (*Demaria et al., 2014*).

## Discussion

Here we uncover a mechanism whereby control of the cell cycle promotes survival and mitogenic signaling in JNK-dependent responses to tissue stress. We demonstrate that JNK signaling induces a dose-dependent extension of G2, which results in either transient stalling or prolonged arrest of cells in G2. Cells in G2 are protected from undergoing JNK-induced apoptosis and promote proliferative signaling to the surrounding tissue in a SASP-like manner (*Figure 8*).

At first sight, some results in our study appeared contradictory. Using *stg*-overexpression to force stalled cells to cycle, we observed (1) apoptosis of JNK-signaling cells in surgically damaged discs, (2) improvement of regenerative capacity in *egr*-expressing discs and (3) reduction of non-autonomous overgrowth in a mosaic tumor model. Our data support a model where any length of G2-stalling protects cells autonomously from JNK-induced apoptosis in G1. Stalled cells switch on a SASP-like phenotype, which when transient supports compensatory growth during regeneration. However, when cells stall chronically, proliferation is autonomously inhibited and chronic SASP-like phenotype drives non-autonomous overgrowth contributing to tumorigenesis. We suggest that our experimental models reveal a spectrum of cell-autonomous (survival, stalling of proliferation) and non-autonomous (mitogenic paracrine signaling) functions of G2-stalling. Importantly, this spectrum is defined by the intensity and length of JNK activity: whereas persistent stalling of tumor cells may cause persistent survival and non-autonomous overgrowth, transient stalling of wound site cells may promote survival transiently and facilitate transient induction of compensatory proliferation.

Surprisingly, both G2-stalling and survival as well as apoptosis directly link to JNK signaling in response to tissue stress. We find that JNK controls *stg/cdc25* transcription and Trbl availability as rate-limiting factors for stalling. Importantly, mouse JNK directly phosphorylates Cdc25C to stall G2/M transitions (*Goss et al., 2003*; *Gutierrez et al., 2010*). If JNK also phosphorylates Stg in flies remains to be determined. As reported before, JNK also regulates *hid* promoting apoptosis (*Shlevkov and Morata, 2012*) and activates Hippo/Yorkie promoting survival by upregulation of *Diap1* (*Sun and Irvine, 2011*; *Sun and Irvine, 2013*). This is analogous to mammalian models where tissue stress induces apoptosis but can also promote cell cycle arrest and survival through upregulation of cell-protective mechanisms. However, the switch between the anti- and pro-apoptotic consequences are not understood (*Fulda et al., 2010*). Importantly, our observations imply that cell cycle progression from G2 to G1 represents the switch between anti- and pro-apoptotic activity of JNK. We suggest that stalling the cell cycle is a protected state, which is dominated by pre-emptive protection from ROS damage through upregulation of the UPR, redox response and other repair pathways. Thus, G2-stalling is important to keep the pro-apoptotic branch of JNK signaling suppressed, a prerequisite of efficient regeneration (*La Fortezza et al., 2016*).

We propose that G2-stalling is induced by JNK itself, and thus only indirectly by cellular damage. While we have not excluded that, for example, proteotoxic stress induces G2-stalling (*Pluquet et al., 2015*), we find that, similar to a previous report (*Wells et al., 2006*), DNA damage is not rate-limiting for cell cycle arrest. In support of this hypothesis, we find that *stg* overexpression rescues architecture and proliferation in *egr*-expressing discs. This indicates that, despite the fact that many JNK-signaling cells die in G1 when bypassing stalling in G2, there are many JNK-signaling G1 cells, which survive, continue to cycle and are healthy enough to contribute to future adult tissues. Even more strikingly, we find that independent of any tissue damage, developmentally

regulated G2-stalling in the wing peripodium is dependent on JNK. Similarly, programmed cell arrest and senescence in mouse embryogenesis is independent of DNA damage or p53, but dependent on a general CDK-inhibitor (p21) and developmental signals of the TGF-β/SMAD or PI3K/FOXO pathways (*Davaapil et al., 2017*; *Muñoz-Espín et al., 2013*). Thus, cell cycle stalling and consequently, SASP or protection from apoptosis, may be under the control of signaling pathways which not always depend on tissue stress. Curiously, another developmentally regulated G2 arrest has already been reported to suppress apoptosis (*Qi and Calvi, 2016*).

At least one previous study has indicated the existence of senescent cells in flies, however their cell cycle stage was less well-defined (*Nakamura et al., 2014*; *Neves et al., 2015*). Many studies have demonstrated correlation of JNK with SASP-like characteristics in flies, such as mitogenic signaling, ECM remodeling or ROS production (*Brock et al., 2017*; *Khan et al., 2017*; *McClure et al., 2008*; *Nakamura et al., 2014*; *Neves et al., 2015*; *Pastor-Pareja and Xu, 2013*; *Ryoo et al., 2004*; *Uhlirova et al., 2005*). We speculate that G2-stalling may either represent a primitive version of a senescent cell cycle arrest that evolved to G0 in mammalian cells or that G2-stalling may also occur in mammals but is less well-defined. Recent studies implicate a role for cells with senescent markers in mammalian wounds and vertebrate development (*Davaapil et al., 2017*; *Demaria et al., 2014*; *Muñoz-Espín et al., 2013*; *Ritschka et al., 2017*) and an injury-induced G2 arrest has been found to interfere with restoration of epithelial homeostasis in a model of chronic kidney disease (*Bonventre, 2014*). Future studies thus need to address if senescence markers can also be found in G2-arrested mammalian cells. G2-stalling may offer an opportunity to protect cells from apoptosis, induce paracrine signals and, importantly, restore active cycling upon restoration of tissue homeostasis, instead of engaging permanent senescence in G0. Thus, more studies are needed to address how G2-stalling may be related to G2 quiescence in stem cells (*Otsuki and Brand, 2018*), to a reversible G2 arrest (*Gire and Dulic, 2015*) or G0 senescence in mammalian tissues.

# Materials and methods

**Key resources table**

| Reagent type (species) or resource | Designation | Source or reference | Identifiers | Additional information |
|---|---|---|---|---|
| Genetic reagent (*Drosophila melanogaster*) | Diap-1-GFP.3.5 | PMID: 18258485 | | |
| Genetic reagent (*D. melanogaster*) | enGAL4, UAS-GFP | | | D.Bilder, UC Berekely |
| Genetic reagent (*D. melanogaster*) | 'eyMARCM19A'; 'ey-FLP, FRT19A tub-GAL80; act5c>y[+]>GAL4, UAS-GFP[S56T]' | PMID: 29494583 | | |
| Genetic reagent (*D. melanogaster*) | 'ubxMARCM82B'; 'Ubx-FLP, tubP-GAL4, UAS-GFP; FRT82B tubP-GAL80' | Bloomington Drosophila Stock Center | BDSC: 42734 | |
| Genetic reagent (*D. melanogaster*) | FRT19A dlg[G0342] | Kyoto Stock Center | DGGR: 111872 | |
| Genetic reagent (*D. melanogaster*) | FRT82B wts[x1] | Bloomington Drosophila Stock Center | BDSC: 44251 | |
| Genetic reagent (*D. melanogaster*) | 'G-TRACE'; 'UAS-FLP.Exel, Ubi-p63E(FRT. STOP)Stinger' | Bloomington Drosophila Stock Center | BDSC: 28282 | |
| Genetic reagent (*D. melanogaster*) | gstD-GFP | PMID: 18194654 | | |

*Continued on next page*

*Continued*

| Reagent type (species) or resource | Designation | Source or reference | Identifiers | Additional information |
|---|---|---|---|---|
| Genetic reagent (*D. melanogaster*) | hep[R75] | Bloomington Drosophila Stock Center | BDSC: 6761 | |
| Genetic reagent (*D. melanogaster*) | 'hid-GFP'; 'hid[MI06452-GFSTF.1]' | Bloomington Drosophila Stock Center | BDSC: 65331 | |
| Genetic reagent (*D. melanogaster*) | mei-41[RT1] | Bloomington Drosophila Stock Center | BDSC: 4169 | |
| Genetic reagent (*D. melanogaster*) | 'puc-LacZ'; 'puc[A251.1F3]' | Bloomington Drosophila Stock Center | BDSC: 11173 | |
| Genetic reagent (*D. melanogaster*) | 'rnGAL4'; 'rn[GAL4-DeltaS]' | Bloomington Drosophila Stock Center | BDSC: 8142 | |
| Genetic reagent (*D. melanogaster*) | 'rn(ts)>'; 'rn[GAL4-DeltaS], tubGAL80[ts]' | Bloomington Drosophila Stock Center | BDSC: 8142; BDSC: 7018 | recombinant |
| Genetic reagent (*D. melanogaster*) | 'rn(ts)>egr'; 'rn[GAL4-5], UAS-egr, tubP-GAL80[ts]' | Bloomington Drosophila Stock Center | | |
| Genetic reagent (*D. melanogaster*) | 'stg-GFP'; 'stg[YD0685]' | Bloomington Drosophila Stock Center | BDSC: 50879 | |
| Genetic reagent (*D. melanogaster*) | 'Trbl-GFP'; 'trbl[MI01025-GFSTF.2]' | Bloomington Drosophila Stock Center | BDSC: 61654 | |
| Genetic reagent (*D. melanogaster*) | TRE-RFP | PMID: 22509270 | | |
| Genetic reagent (*D. melanogaster*) | UAS-bsk[DN] | Bloomington Drosophila Stock Center | BDSC: 6409 | |
| Genetic reagent (*D. melanogaster*) | UAS-Cdk1 RNAi [TRiP.HMS01531] | Bloomington Drosophila Stock Center | BDSC: 36117 | |
| Genetic reagent (*D. melanogaster*) | UAS-GFP S56T | | BDSC: 1521 | |
| Genetic reagent (*D. melanogaster*) | UAS-grp RNAi [TRiP.HMC05162] | Bloomington Drosophila Stock Center | BDSC: 62155 | |
| Genetic reagent (*D. melanogaster*) | UAS-hep[act] | Bloomington Drosophila Stock Center | BDSC: 9306 | |
| Genetic reagent (*D. melanogaster*) | UAS-hid | | | G. Morata, CBSMO Spain |
| Genetic reagent (*D. melanogaster*) | UAS-mei-41 RNAi [TRiP.GL00284] | Bloomington Drosophila Stock Center | BDSC: 35371 | |
| Genetic reagent (*D. melanogaster*) | UAS-stg | Bloomington Drosophila Stock Center | BDSC: 4777 | |

*Continued on next page*

*Continued*

| Reagent type (species) or resource | Designation | Source or reference | Identifiers | Additional information |
|---|---|---|---|---|
| Genetic reagent (*D. melanogaster*) | UAS-stg.HA | Bloomington Drosophila Stock Center | BDSC: 56562 | |
| Genetic reagent (*D. melanogaster*) | UAS-Xbp1-GFP.HG | Bloomington Drosophila Stock Center | BDSC: 60731 | |
| Genetic reagent (*D. melanogaster*) | 'E2F1-FUCCI, CycB-FUCCI'; 'Ubi-GFP.E2f1.1–230, Ubi-mRFP1.NLS.CycB.1–266' | Bloomington Drosophila Stock Center | BDSC: 55123 | |
| Genetic reagent (*D. melanogaster*) | 'CycB-FUCCI'; 'Ubi-mRFP1.NLS.CycB.1–266' | PMID: 24726363 | | |
| Genetic reagent (*D. melanogaster*) | 'ubx-flp;; FRT82B ubi-GFP' | | | I. Hariharan, UC Berkeley |
| Genetic reagent (*D. melanogaster*) | upd-lacZ (PD) | PMID: 8582614 | | |
| Genetic reagent (*D. melanogaster*) | UAS-trbl RNAi [TRiP.HMJ02089] | Bloomington Drosophila Stock Center | BDSC: 42523 | |
| Genetic reagent (*D. melanogaster*) | UAS-P35 | Bloomington Drosophila Stock Center | BDSC: 5072 | |
| Genetic reagent (*D. melanogaster*) | GH146-QF, QUAS-mtdTomato-3xHA | Bloomington Drosophila Stock Center | BDSC: 30037 | |
| Genetic reagent (*D. melanogaster*) | Ubi-CasExpress | Bloomington Drosophila Stock Center | BDSC: 65419 | |
| Antibody | Rabbit anti-cleaved Dcp-1 | Cell Signaling | Cat. #: 9578 | (1:200) |
| Antibody | Mouse monoclonal anti-CycB | Developmental Studies Hybridoma Bank | Cat. #: F2F4 | (1:20) |
| Antibody | Rat monoclonal anti-DE-cadherin | Developmental Studies Hybridoma Bank | Cat. #: DCAD2 | (1:100) |
| Antibody | Mouse monoclonal anti-discs large | Developmental Studies Hybridoma Bank | Cat. #: 4F3 | (1:100) |
| Antibody | Mouse anti-β-Galactosidase | Promega | Cat. #: Z3782 | (1:1000) |
| Antibody | Chicken anti-GFP | Abcam | Cat. #: ab13970 | (1:1000) |
| Antibody | Rabbit monoclonal anti-GFP | Invitrogen | Cat. #: G10362 | (1:200) |
| Antibody | Rabbit anti-H2Av-pS137 | Rockland | Cat. #: 600-401-914 | (1:500) |
| Antibody | Mouse anti-H3-pS10 | Abcam | Cat. #: ab14955 | (1:2000) |
| Antibody | Rat monoclonal anti-HA | Monoclonal Antibody Core Facillity at the Helmholtz Zentrum München | Clone #: 3F10 | (1:20) |
| Antibody | Mouse monoclonal anti-MMP1 | Developmental Studies Hybridoma Bank | Cat. #: 3A6B4 | (1:30) |
| Antibody | Mouse monoclonal anti-MMP1 | Developmental Studies Hybridoma Bank | Cat. #: 3B8D12 | (1:30) |

*Continued on next page*

*Continued*

| Reagent type (species) or resource | Designation | Source or reference | Identifiers | Additional information |
|---|---|---|---|---|
| Antibody | Mouse monoclonal anti-MMP1 | Developmental Studies Hybridoma Bank | Cat. #: 5H7B11 | (1:30) |
| Antibody | Mouse monoclonal anti-RFP | Abcam | Cat. #: ab65856 | (1:100) |
| Antibody | Rat monoclonal anti-RFP | Monoclonal Antibody Core Facillity at the Helmholtz Zentrum München | Clone #: 5F8 | (1:20) |
| Antibody | Mouse monoclonal anti-Ultrabithorax | Developmental Studies Hybridoma Bank | Cat. #: Ubx FP3.38 | (1:10) |
| Commercial assay or kit | Click-iT Plus EdU Alexa Fluor 647 Imaging Kit | Invitrogen | Cat. #: C10640 | |

### *Drosophila* stocks and genetics

Flies were kept on standard food and raised at 18°C and 30°C (expression of pro-apoptotic trans-genes), or 25°C. A list of strains, detailed genotypes and experimental conditions are provided in *Supplementary file 1* and in the key resources table.

### Tissue injury models

#### In situ surgical wounding

Wounding of wing imaginal discs in situ was performed by quickly immobilizing L3 larvae in ice-cold PBS and applying pressure to the fluorescently-labeled wing imaginal disc with a 0.125 mm tungsten needle (Fine Science Tools, 10130–05) without perforating the larval cuticle, as described by *Bryant (1971)*; *Yoo et al. (2016)*. After wounding, the larvae were immediately placed in a new food vial and allowed to recover at 25°C for the indicated time. For each animal, only the right wing disc was wounded. The left wing disc was used as undamaged control. For each experiment involving functional genetics, a wild type control was included, and the wounding procedure was performed in blind. At least 15 larvae were wounded for each genotype.

#### Temporal and spatial control of pro-apoptotic transgene expression

To induce expression of *egr* or *hid*, experiments were carried out as described in *Smith-Bolton et al. (2009)* and *La Fortezza et al. (2016)* with few modifications. Briefly, larvae of geno-type *rnGAL4, tub-GAL80*$^{ts}$ (denoted as *rn*$^{ts}$>) and carrying the desired *UAS-transgenes* were staged by a 6 hr egg collection and raised at 18°C at the density of 50 larvae/vial. Overexpression of trans-genes was induced by shifting the temperature to 30°C for 24 hr at day seven after egg deposition (AED) (*Figure 2—figure supplement 1A*). Larvae were subsequently allowed to recover at 18°C for the indicated time (recovery time R24-R72 hours, or adulthood), or dissected immediately (R0). All images are R0, unless noted otherwise. Control genotypes were either *rn*$^{ts}$>, or the siblings of the ablating animals (*+/TM6B, tubGAL80*) (*Smith-Bolton et al., 2009*). At least 16 discs were dissected for each genotype per replicate.

#### Mosaic tumor models

To obtain MARCM clones in the wing discs, larvae cultures were synchronized by a 6 hr egg collec-tion and raised at 25°C at a density of 50 larvae/vial. Control larvae were analyzed 5 days AED, while larvae carrying *wts*$^{x1}$ clones were analyzed 7 days AED, to account for developmental delay induced by the presence of tumorigenic cells. To obtain MARCM clones in the eye discs, equal crosses for all genotypes were set up in parallel and processed 6 days AED. Discs expressing and stained for HA-tagged Stg were mounted on the same slide as their respective control discs, to ensure comparabil-ity of volume quantifications.

## Flow cytometry

Cell cycle analysis of wing imaginal discs by flow cytometry was performed as described (*de la Cruz and Edgar, 2008*). Wing imaginal discs from at least 10 larvae were dissected in PBS and incubated for 2 hr in PBS containing 9X Trypsin-EDTA (Sigma, T4174) and 0.5 µg/ml Hoechst 33342 (Invitrogen, H3570). Cells were analyzed with an LSRFortessa cell analyzer (BD Biosciences) or FACS Aria II cell sorter (BD Biosciences). Univariate cell cycle analysis was performed using the Watson Pragmatic algorithm in FlowJo v10 (FlowJo).

## Immunohistochemistry

### General comments

Where possible, control and experimental samples were fixed, processed and mounted together to ensure comparable staining and imaging conditions. The signals of the following fluorescent reporters were further amplified by anti-GFP or anti-RFP antibody staining: CycB-FUCCI, E2F1-FUCCI, G-TRACE, Hid-GFP, *stg*-GFP, Trbl-GFP, Xbp1-GFP.HG.

### Immunohistochemistry

Larvae were dissected, and cuticles were fixed for 15 min at room temperature in 4% paraformaldehyde. Washing steps were performed in 0.1% Triton X-100/PBS (PBT), blocking in 5% NGS/PBT. The following antibodies were incubated overnight at 4°C: rabbit anti-cleaved Dcp-1 (Cell Signaling, 9578, 1:200), mouse anti-Cyclin B (DSHB, F2F4, 1:20), rat anti-DE-cadherin (DSHB, DCAD2, 1:100), mouse anti-discs large (DSHB, 4F3, 1:100), mouse anti-β-Galactosidase (Promega, Z3783, 1:1000), chicken anti-GFP (Abcam, ab13970, 1:1000), rabbit anti-GFP (Invitrogen, G10362, 1:200), rabbit anti-H2Av-pS137 (Rockland, 600-401-914, 1:500), mouse anti-H3-pS10 (Abcam, ab14955, 1:2000), rat anti-HA (MAB facility at the Helmholtz Zentrum München, 3F10, 1:20), mouse anti-MMP1 (DSHB, a mix of 3A6B4, 3B8D12 and 5H7B11, each 1:30), mouse anti-RFP (Abcam, ab65856, 1:100), rat anti-RFP (MAB facility at the Helmholtz Zentrum München, 5F8, 1:20), mouse anti-Ultrabithorax (DSHB, FP3.38, 1:10). EdU incorporation was performed for 15 min, unless noted otherwise, and detected using the Click-iT Plus EdU Alexa Fluor 647 Imaging Kit (Invitrogen, C10640) prior to primary antibody incubation, according to manufacturer's guidelines. Tissues were counterstained with DAPI (0.25 ng/µl, Sigma, D9542) during incubation with cross-adsorbed secondary antibodies coupled to Alexa Fluorophores (Invitrogen or Abcam) at room temperature for 2 hr. Tissues were mounted using SlowFade Gold Antifade (Invitrogen, S36936). Samples were imaged using Leica TCS SP5 or SP8 confocal microscopes.

## Image analysis and quantification

### General comments

Images were processed, analyzed and quantified using tools in Fiji (ImageJ v2.0.0) (*Schindelin et al., 2012*). Figure panels were assembled using Photoshop CS5 (Adobe). Statistical analyses were performed in R v3.3.3 (www.R-project.org).

### Peripodial cell cycle quantification

A mask of the DAPI counterstain, obtained from total projections of confocal stacks containing the entire wing discs, was used to estimate the total disc size by measuring its area with the 'Analyze Particles' tool in Fiji. To analyze a defined population of cells in the peripodium of size-matched wing discs, Ubx +nuclei were identified with 'Analyze Particles' (size = 10.00–40.00 circularity = 0.50–1.00), after applying 'Unsharp Mask' (radius = 10 mask=0.4), Gaussian Blur (sigma = 0.2 scaled) and Watershed functions. The nuclei of the resulting mask were counted and displayed as total Ubx +cells. Automated determination of the cell cycle phase for each nucleus was obtained by measuring the average fluorescence intensity of both FUCCI reporters. The criteria to define each cell cycle phase are shown in *Figure 3—figure supplement 1E*.

### TRE-RFP quantification

To measure the fluorescence intensity of *TRE*-RFP signals in injured, *egr*-expressing or tumor discs, a small circular ROI of fixed radius (25–30 µm) was placed in an area of high *TRE*-RFP signal of a single

confocal section for each stack, carefully chosen to capture maximal JNK activity in the disc proper, avoiding the peripodium and extruded cell debris. The mean intensity of each ROI was obtained using the 'Measure' function in Fiji.

### Dcp-1 quantification in injured discs
Masks of Dcp-1 signals were obtained in Fiji from maximum intensity projections of confocal stacks by applying a fixed threshold (value = 75) and the 'Remove outliers' function (bright, radius = 1.5). Areas of the resulting masks were obtained using 'Analyze Particles'.

### FUCCI and Dcp-1 intensity analysis in egr-expressing discs
A single confocal section for each stack was carefully chosen within the tissue to capture apoptotic cells before extrusion. An ROI corresponding to the pouch was selected using the 'Freehand Selection' tool in Fiji. Pixel fluorescence intensities for all channels were subsequently obtained using the 'Save XY Coordinates' function, after applying a 'Gaussian Blur' filter (radius = 0.3 μm) to reduce noise. Data from $n$ = 5 (*Figure 6G*) and $n$ = 4 discs (*Figure 6I*) were pooled and the distribution of fluorescence intensities of each pixel was represented as 2D density plots.

### GFP volume and total disc size quantification (mosaic clone analysis)
Masks for GFP signals (positively labeled clones) and total volume were obtained in Fiji from entire confocal stacks by applying the functions 'Auto Threshold' (settings: 'Li' for GFP or 'Triangle' for total, white objects, stack histogram) and 'Remove Outliers' (settings: black and white pixels removal, radii = 2) to GFP signals or to the sum of DAPI and GFP signals, respectively. The resulting masks were analyzed using the '3D Objects Counter' function (settings: threshold = 128, min = 50 max=Inf). The sum of the resulting object volumes for each disc was used to describe the GFP and total volume of the disc. Non-GFP volumes were calculated by subtracting the GFP volume from the total disc volume.

### Mitotic density
PH3-positive cells were counted in Fiji by applying 'Auto threshold' (settings: 'Yen') and 'Remove Outliers' (settings: black and white pixels removal, radii = 1) to single confocal sections. A total of 5 equally spaced (6 μm) confocal sections for each of 6 stacks per sample were analyzed. Masks for GFP signals were obtained by applying 'Auto threshold' (settings: 'Triangle) and 'Remove Outliers' (settings: dark, radius = 3; bright, radius = 1). GFP-negative areas were calculated by subtracting GFP-positive areas from the total tissue area.

### **Adult wing size analysis**
Adult flies were collected 12 hr after eclosion and stored in 2-propanol. Wing sizes were indexed by binning into five different wing phenotypic classes: 0%, 25%, 50%, 75% or 100% of wild type size, as defined in *La Fortezza et al. (2016)*; *Smith-Bolton et al. (2009)*. Importantly, wings smaller than 100% of wild type size, typically present a range of morphological defects (*Smith-Bolton et al., 2009*). Of note, wings of 100% size but with notches or incomplete vein formation were classified as 75%.

## Acknowledgements
We thank the reviewers for critical comments on the manuscript. We thank the CALM facility at LMU and the LIC facility at the University of Freiburg for technical help with imaging. Flow cytometry analysis was performed with the help of S Bultmann (LMU München) and V Hilgers (MPI-IE, Freiburg). We thank D Bohmann, B Edgar, K Irvine, G Morata, YH Sun and N Zielke for sharing reagents. We thank the Bloomington Drosophila Stock Center (BDSC), the Kyoto Stock Center (DGGR), the MAB facility at the Helmholtz Zentrum München and the Developmental Studies Hybridoma Bank (DSHB) for providing fly stocks and antibodies. We thank the IMPRS-LS and SGBM graduate schools for supporting our students. Funding for this work was provided by the DFG (CL490/1) and Germany's Excellence Strategy (CIBSS – EXC-2189 – Project ID 390939984), as well as by the Boehringer Ingelheim Foundation (Plus3).

# Additional information

## Funding

| Funder | Grant reference number | Author |
| --- | --- | --- |
| Deutsche Forschungsgemeinschaft | CL490-1/1 | Anne-Kathrin Classen |
| Boehringer Ingelheim Stiftung | Plus3 Programme | Anne-Kathrin Classen |
| Deutsche Forschungsgemeinschaft | EXC-2189 - Project ID 390939984 | Anne-Kathrin Classen |

The funders had no role in study design, data collection and interpretation, or the decision to submit the work for publication.

## Author contributions

Andrea Cosolo, Conceptualization, Data curation, Formal analysis, Validation, Investigation, Visualization, Methodology, Writing—original draft, Writing—review and editing; Janhvi Jaiswal, Gábor Csordás, Data curation, Formal analysis, Validation, Investigation, Visualization, Writing—review and editing; Isabelle Grass, Validation, Investigation, Methodology; Mirka Uhlirova, Supervision, Writing—review and editing; Anne-Kathrin Classen, Conceptualization, Formal analysis, Supervision, Funding acquisition, Visualization, Writing—original draft, Writing—review and editing

## Author ORCIDs

Andrea Cosolo (iD) http://orcid.org/0000-0003-3417-0713
Gábor Csordás (iD) http://orcid.org/0000-0001-6871-6839
Mirka Uhlirova (iD) https://orcid.org/0000-0002-5735-8287
Anne-Kathrin Classen (iD) http://orcid.org/0000-0001-5157-0749

## Decision letter and Author response

Decision letter https://doi.org/10.7554/eLife.41036.028
Author response https://doi.org/10.7554/eLife.41036.029

# Additional files

## Supplementary files

• Supplementary file 1. Genotypes and experimental conditions. This table lists detailed genotypes and temperature conditions used to generate the data for each of the main and supplementary figures in this study.
DOI: https://doi.org/10.7554/eLife.41036.019

• Transparent reporting form
DOI: https://doi.org/10.7554/eLife.41036.020

## Data availability

All data generated or analysed during this study are included in the manuscript and supporting files.

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
