## [Decision Letter]

Thank you for submitting your article "JNK-dependent cell cycle stalling in G2 promotes survival and senescence-associated signaling in tissue stress" for consideration by *eLife*. Your article has been reviewed by three peer reviewers, and the evaluation has been overseen by Andrea Musacchio as the Senior and Reviewing Editor. The reviewers have opted to remain anonymous.

The reviewers have discussed the reviews with one another and the Reviewing Editor has drafted this decision to help you prepare a revised submission.

Summary:

The manuscript reports a study on the role of JNK signalling in different situations where this stress-related cascade is activated in *Drosophila* imaginal discs. JNK signalling is well known to be involved in disc tissue damage responses that contribute to apoptosis, regeneration and tumour formation. How JNK signalling can produce such different outcomes is a long-standing question. Studying different JNK signalling contexts in discs, the authors propose that JNK signalling induces transient arrest of cells in G2, which the authors call G2-stalling, under a variety of insults. This transient arrest in G2 protects cells from JNK-induced apoptosis and potentiates senescent-like phenotypes of these cells. G2 stalled cells may also be able to signal to neighbouring cells influencing also their behaviour. The reviewers agree that the study identifies G2 arrest as a critical aspect of the response of tissues to JNK signalling, and that this contributes to understanding tissue damage, regeneration and different outcomes of JNK signalling. The reviewers, however, also identify several points that require further attention. Please note that some of the points raised are closely related, but kept separate as they were raised by different reviewers.

Essential revisions:

1) The authors identify a possible mechanistic link between JNK signaling and G2 arrest in the upregulation of tribbles, a known negative regulator of G2-M transition. This seems really clear in Figure 4—figure supplement 1, showing striking tribbles upregulation in wounded discs. Why is this shown only in a supplementary figure? This is in my opinion one of the main findings in the paper, potentially taking it to a truly mechanistic level. Can the authors show induction of Tribbles is JNK-dependent? Can the authors show functional involvement of tribbles in JNK-induced G2 arrest?

2) Regarding the data showing G2 arrest in mosaic tumors (Figure 6), it seems very surprising the G2 shift in *wts*- clones, as previous analysis of other pathway mutants showed faster cycling but wt FACS cell cycle profiles, with no G2 shift (Tapon et al., 2002; Harvey et al., 2003). The authors state that the G2 shift is specific to TRE-positive (high JNK) cells, but the FACS profile in 6B' shows otherwise. It is important to make sure that this G2 shift is really a *wts* phenotype, and not, for instance, due to additional mutations in the FRT chromosome. For this, clones for other hippo mutants or *yki*-overexpression should be analysed. If real, the discrepancy with previous studies of Hippo/cell cycle should be acknowledged. If not real, these *wts* data could be removed to focus on the *dlg*- clones (Figure 7—figure supplement 1), showing the adult eyes are actually smaller when *dlg*- clones express string and that *upd* induction is reduced.

3) Regarding a normal role of JNK signaling promoting G2 arrest in the peripodial epithelium, this is again an important finding hidden in supplemental information, making this reviewer wonder how confident the authors are in their own data. To confirm JNK signaling is arresting cells in G2 in the peripodial epithelium: how did the authors define the peripodial epithelium to perform their counts? Is the peripodial epithelium in hepr75 (JNKK) discs larger? Does it show more EdU staining compared to S7F'? More PH3+ cells?

4) The finding that the proportion of cells in G2 correlates with JNK activity induced by injury, genetic cell ablation or loss of tumour suppressor genes is convincing. JNK activity is necessary and sufficient to induced G2 markers, which is convincingly supported by data. However, the authors may wish to consider the following points. Some concepts already established and linked to JNK need to be considered/discussed more clearly in the text. What about DNA damage response? Cdc25 phosphatase activity? Regulation or cell competition?

5) The idea of testing different JNK inducing conditions in terms of duration is very interesting and important. While it appears very likely that JNK signaling is different in these conditions, it would be important to show that. To that end, it might be desirable to provide TRE-reporter time-course for injury, *egr* RNAi and *wts* clones, showing that the TRE response curves are indeed different. Also, is it well established that the TRE reporter faithfully reports on JNK signaling dose in vivo, how is/was that shown? The scaling of TRE signal and G2 increase should be detectable by FACS. Further relevant here, is TRE signal affected in *egr stg* double RNAi pouches? It would be important to show that in the double RNAi TRE signal is still similar to *egr* RNAi alone.

6) The idea that JNK impinges on the cell cycle seems plausible. The study evokes Cdc25 downregulation through Tribbles upregulation as a potential mechanism as how JNK signalling could impinge on the cell cycle. Surprisingly there is no mentioning of literature that JNK can phosphorylate Cdc25 in HEK293 cells attenuating its phosphatase activity (e.g. Gutierrez et al., 2010), therefore there is a direct link on Cdc25 activity regulation, without the need to downregulate Cdc25. The GFP trap line for string is apparently not homozygous viable and hence not functional, so I am not convinced that the data show that *stg* levels actually decline in G2 stalled cells.

7) JNK is tightly linked to the G2/M DNA damage response checkpoint. Can it be ruled out that this checkpoint is triggered in the conditions used? Pushing cells through the cell cycle by Cdc25 overexpression can override DNA damage response checkpoints in certain systems. It would be important to test that injury, temporal *egr* RNA and *wts* do not activate the DNA damage response (DDR). There are some important experiments in Supplementary Figure 7 in the manuscript in this regard, buy they are oddly placed and discussed. Is the reported G2 stalling clearly distinguishable from G2 cell cycle arrest induced by DNA damage?

8) While the non-autonomous activation of TRE is clear and also previously reported, the situation could be more complex. String overexpression might affect cell fitness and therefore cell competition could be an issue here. Can this be clearly separated, ruled out? That would be important to address.

9) A more general concern is that novel bits seemed investigated in less detail than the already well established findings. The non-cell autonomous influence of tumour cells on neighbours to direct faster growth is fascinating but not well explored, and the suggestion that stalled cells were transiently senescent and releasing SARPs is too rushed to be definitive. Nonetheless, this is highlighted in the title and Abstract of this paper. Finally, are G2-stalled cells really protected from apoptosis? It seems that both pro- and anti-apoptotic pathways are activated; is knockdown of Hippo/Yorkie signalling in these cells needed?

---

## [Author Response]

Essential revisions:1) The authors identify a possible mechanistic link between JNK signaling and G2 arrest in the upregulation of tribbles, a known negative regulator of G2-M transition. This seems really clear in Figure 4—figure supplement 1, showing striking tribbles upregulation in wounded discs. Why is this shown only in a supplementary figure? This is in my opinion one of the main findings in the paper, potentially taking it to a truly mechanistic level. Can the authors show induction of Tribbles is JNK-dependent? Can the authors show functional involvement of tribbles in JNK-induced G2 arrest?

Our previous analysis has focused on Stg, as our experiments suggested that Stg overexpression was sufficient to suppress G2-stalling. However, we are now providing this new data in the manuscript:

1) Tribbles-GFP is also upregulated by acute surgical damage, and not only in *egr-*expressing discs (anterior compartment in Figure 4E).

2) Tribbles-GFP upregulation depends on JNK activity, because expression of a dominant-negative *Bsk^DN^* using the *enGAL4* driver prevents high Trbl-GFP expression in the posterior compartment of surgically damaged discs (posterior compartment in Figure 4E).

3) A UAS-*trbl* RNAi line, which efficiently knocks down Trbl-GFP in wing discs (Author response image 1), has little effect on normal wing development (Figure 4—figure supplement 1L). However, *trbl* RNAi efficiently suppresses G2-stalling in *egr-*expressing discs and restores a heterogenous cell cycle profile (Figure 4H).

Combined, these new data suggest that Tribbles, like Stg, is a rate-limiting effector of JNK for G2-stalling. Thus, it is conceivable that Tribbles may regulate G2-stalling also by regulating proteasomal degradation of Stg protein.

Importantly, however, we conclude from the analysis of the *stg-GFP* trap line that *stg* is at least transcriptionally downregulated in G2 stalled cells, a mechanism likely unrelated to Tribbles function, which is expected to act on Stg proteins. Since we have been unable to detect Stg protein levels directly (for example an antibody recognizing Stg in embryos (Di Talia et al., 2013) does not detect Stg in imaginal discs, possibly because of different fixation conditions – see discussion to point 4), we cannot make any statement about how Stg protein levels may be regulated in response to tissue damage, JNK or in response to Tribbles upregulation.

Based on the literature, Tribbles is expected to also regulate many proteins related to cell growth, cell migration and metabolism (for example (Das et al., 2014; Masoner et al., 2013). Because Stg is a more direct effector of G2/M transitions, we have decided to focus our functional studies on UAS-stg overexpression.

2) Regarding the data showing G2 arrest in mosaic tumors (Figure 6), it seems very surprising the G2 shift in wts- clones, as previous analysis of other pathway mutants showed faster cycling but wt FACS cell cycle profiles, with no G2 shift (Tapon et al., 2002; Harvey et al., 2003). The authors state that the G2 shift is specific to TRE-positive (high JNK) cells, but the FACS profile in 6B' shows otherwise. It is important to make sure that this G2 shift is really a wts phenotype, and not, for instance, due to additional mutations in the FRT chromosome. For this, clones for other hippo mutants or yki-overexpression should be analysed. If real, the discrepancy with previous studies of Hippo/cell cycle should be acknowledged. If not real, these wts data could be removed to focus on the dlg- clones (Figure 7—figure supplement 1), showing the adult eyes are actually smaller when dlg- clones express string and that upd induction is reduced.

We completely agree with the reviewer that previous studies (Harvey et al., 2003; Huang et al., 2005; Tapon et al., 2002) have unequivocally demonstrated that Hpo/Sav/*Wts* pathway mutants cycle faster and have normal cell cycle profiles. To address this independently, we analyzed *yki^CA^ (yki^S168A.V5^*) -overexpressing mosaic wing discs (Worley et al., 2013) by flow cytometry analysis (see Author response image 2), confirming that cell cycle profile of *yki^CA^*-overexpressing cells overlaps perfectly with wild type cells. However, TRE-positive wild type and TRE-positive *yki^CA^*-overexpressing cells (both about 12% of the total population) display a pronounced shift towards G2, strongly supporting our conclusion that JNK-activity in any genotype induces G2 stalling. Of note, JNK upregulation in *yki^CA^*-overexpressing is not extensive enough to average into significant cell cycle changes visible in flow cytometry analysis of the entire cell population, also explaining why this shift has not been described in previous studies.

Thus, we agree that cell cycle changes in TRE-negative *wts^x1^* mutant cells could be due to a second site mutation on that chromosome.

**Author response image 2. respfig2:** 

Importantly, however, we are not drawing any conclusions about cell cycle changes in TRE-negative *wts^x1^* mutant cells, as these are not at all subject of our study. However, we chose to work with this allele because many tumor suppressor mutations that activate JNK become eliminated from mosaic discs and are thus challenging to study by flow cytometry (for example *scrib/dlg/lgl* mutations). However, *wts^x1^* cells form large clones and display robust JNK activation, making their analysis robust. In fact, analysis of mosaic *hpo^42-47^*and *yki^CA^*-overexpressing disc showed much lower and more localized levels of JNK activity (Author response image 3)

**Author response image 3. respfig3:** 

As we do see an even stronger G2 shift in TRE-positive than in TRE-negative *wts^x1^* cells, we conclude that this additional shift is due to JNK activity. Because we verify that *yki^CA^* and *scrib/dlg/lgl* cells stall in G2, we suggest that combined the analysis of all genotypes support our point that tumor suppressor or oncogenic mutations which activate TRE also stall in G2.

We previously explored the idea to present adult eyes of *dlg1^G0342^*and *dlg1^G0342,^ UAS-stg* genotypes. In fact, preliminary experiments suggested that the adult eyes of *dlg1^G0342,^ UAS-stg* were smaller than in *dlg1^G0342^*alone, consistent with the idea that mitogenic signaling may be impaired. However, this interpretation was limited by possible alternative interpretations related to *dlg1^G0342,^ UAS-stg* cells failing to differentiate and contribute to adult eye tissues during pupal morphogenesis. We thus decided to not further explore this data at this point in time and to focus on imaginal disc phenotypes. In fact, we are now better experimentally addressing non-autonomous mitogenic behavior in Figure 7 and our response to point 8.

Based on these arguments and the extensive old and new data available for *wts^x1^*, we are strongly favoring to focus on the *wts^x1^* data in the manuscript.

To acknowledge the abnormal behavior of TRE-negative *wts^x1^* cells, we added the following to Figure 7 in the legend: “Of note, previous cell cycle studies of Hippo-pathway mutant have not reported any alterations (Harvey et al., 2003; Huang et al., 2005; Tapon et al., 2002). Therefore, the mild cell cycle shift in TRE-negative *wts^x1/x1^* cells appears to be specific to the *wts^x1^*allele.”

3) Regarding a normal role of JNK signaling promoting G2 arrest in the peripodial epithelium, this is again an important finding hidden in supplemental information, making this reviewer wonder how confident the authors are in their own data. To confirm JNK signaling is arresting cells in G2 in the peripodial epithelium: how did the authors define the peripodial epithelium to perform their counts? Is the peripodial epithelium in hepr75 (JNKK) discs larger? Does it show more EdU staining compared to S7F'? More PH3+ cells?

We are very confident about the data as we have repeated this experiment multiple times. During the preparation of the manuscript we received feed-back that describing the peripodium in the main results extends the first part of the manuscript at the expense of the important functional data for G2-stalling in the second half. However, we absolutely agree with the reviewer that this is a crucial point in the paper.

We have again repeated our analysis but used Ubx expression as a marker for cells localized in the central peripodial domain (a few TRE positive G2 cells are also located anterior of this domain) and now represent this analysis in Figure 3D-G and Figure 3—figure supplement 1A-E:

We included wild type and *hep^R75^* discs of the same size range in our analysis. These discs had statistically similar counts of Ubx-positive cells. We further improved the cell cycle analysis by using an automated and unbiased approach in FIJI, to measure the fluorescence intensity of the FUCCI reporters in each Ubx-positive cell and thus identify the cell cycle phase. This analysis confirms that there are significant differences in the number of G2 and G1 cells between wild type and *hep^R75^* peripodia. S-phase counts were not statistically significantly different. Combined we conclude, that even though *hep^R75^*discs may have less G2 cells in Ubx-positive domains, cell division rates may not be abnormally increased. This is actually consistent with data by the Edgar lab suggesting that the duration of individual cell cycle phases is intrinsically compensated to maintain the cell cycle length constant (Reis and Edgar, 2004).

4) The finding that the proportion of cells in G2 correlates with JNK activity induced by injury, genetic cell ablation or loss of tumour suppressor genes is convincing. JNK activity is necessary and sufficient to induced G2 markers, which is convincingly supported by data. However, the authors may wish to consider the following points. Some concepts already established and linked to JNK need to be considered/discussed more clearly in the text. What about DNA damage response? Cdc25 phosphatase activity? Regulation or cell competition?

As above, during the preparation of the manuscript we received feedback that describing the DNA damage data early on extends the first part of the manuscript at the expense of the important functional data for G2-stalling. As we absolutely agree with the reviewer that DNA damage is a crucial concept in cell cycle arrests, we have moved the data to Figure 3—figure supplement 1. We choose to present it in the supplements because our observations align with a previously published paper by the Johnston lab (Wells et al., 2006), who similarly observed that DNA damage is not involved in the regulation of a cell cycle arrest in undead cells (which we think represents the same phenomenon). We now emphasize in the text that JNK signaling is sufficient for G2-stalling but that it may be a downstream effector of cellular damage signals, such as heavy DNA damage or proteotoxic stress. Importantly, however, as JNK signaling during development (which would be independent of a (DNA) damage context) is sufficient for G2-stalling, we conclude that DNA damage could be an upstream regulator (via activation of JNK) but that itself is not necessary for stalling if other JNK-activating signals dominate.

Because of the literature on mammalian Cdc25c being phosphorylated by JNK to drive its proteasomal degradation (Goss et al., 2003; Gutierrez et al., 2010), we initiated studies to look at String protein levels using IF and Western blots, and to analyze String phosphorylation levels using Phospho-tag gel shift assays in different genetic backgrounds in vivo. However, we encountered several technical challenges: the *stg*-GFP trap does not give rise to a functional fusion protein, an HA-tagged Stg protein (FBrf0225451) is only detected in a very short window of the cell cycle (late G2/M) and thus hard to visualize on Western blots, and an antibody recognizing Stg in embryos (Di Talia et al., 2013) does not detect Stg in imaginal discs (possibly because of different fixation conditions). We thus have stopped this line of investigation until we have more resources to investigate Stg protein levels, modifications and activity in imaginal disc.

However, we included the references to the mammalian data (Goss et al., 2003; Gutierrez et al., 2010) now in the Discussion because we agree that these studies align with our observations.

We address cell competition in our response to point 8.

5) The idea of testing different JNK inducing conditions in terms of duration is very interesting and important. While it appears very likely that JNK signaling is different in these conditions, it would be important to show that. To that end, it might be desirable to provide TRE-reporter time-course for injury, egr RNAi and wts clones, showing that the TRE response curves are indeed different. Also, is it well established that the TRE reporter faithfully reports on JNK signaling dose in vivo, how is/was that shown? The scaling of TRE signal and G2 increase should be detectable by FACS. Further relevant here, is TRE signal affected in egr stg double RNAi pouches? It would be important to show that in the double RNAi TRE signal is still similar to egr RNAi alone.

We completely agree that the temporal resolution of JNK levels is crucial to the interpretation of the data and have now integrated the following new data in the manuscript and in a new model summary in Figure 8:

1) a comparison of TRE-RFP levels in surgically injured disc, *egr-*expressing discs and *wts^X1^* mosaic tumor discs, demonstrating that *egr-*expressing discs have extreme and abnormally high levels of JNK activity if compared to surgical injury or *wts^X1^*mosaic tumor discs (Figure 7B-D)

2) a quantification of the decline in TRE-RFP fluorescence intensity after surgical injury and *egr-*induced tissue damage demonstrating that JNK signaling after surgical injury declines faster than after *egr-*expression (Figure 5M-N).

3) In addition, we demonstrate that TRE-RFP levels in *wts^X1^*mosaic tumor discs increases during larval stages and stays chronically high (Figure 7—figure supplement 1A).

With respect to TRE reporter activity:

1) The scaling of the TRE signal and G2 increase detected by FACS analysis was already included in the first manuscript version in Figure 2L. We modified the visualization of the data to emphasize this point further.

2) We now also demonstrate that the TRE-reporter is equally active in *egr* and *egr,stg* expressing imaginal discs, confirming that JNK activity is not affected by *stg*-coexpression (Figure 5—figure supplement 1H).

We furthermore tested the ability of the TRE reporter to sensitively detect JNK activity dose in vivo.

1) TRE-activity completely co-localizes with other established JNK-effector read-outs, such as MMP1 expression (Uhlirova and Bohmann, 2006) (Author response image 4), suggesting that TRE is a faithful JNK-reporter.

**Author response image 4. respfig4:** 

2) Expression of a strong dominant-negative Bsk^DN^ construct in the posterior domain using *enGAL4* (Author response image 5) completely abrogates TRE-RFP activation in the posterior compartment of surgically injured discs (white arrowheads). This recapitulates repression of *puc*-LacZ in the same context (Figure 3B) and confirms that JNK is the major activator of the TRE reporter.

**Author response image 5. respfig5:** 

3) Similarly, co-expression of a dominant-negative Bsk^DN^ construct in *egr-*expressing cells (Author response image 6) strongly suppresses TRE-RFP activation in *egr-*expressing discs. In contrast, *egr-*expressing discs hemizygous for *hep^R75^* allele have only reduced TRE-reporter intensity. *hep^R75^* is a short deletion around the ATG translation initiation start site (Glise et al., 1995). As the TSS site is theoretically still intact, this deletion could be bypassed by a downstream in-frame ATG that localizes just upstream of the kinase domain, rendering the *hep^R75^* allele a hypomorph. Thus, the TRE reporter responds sensitively to different levels of JNK activity in vivo.

**Author response image 6. respfig6:** 

We hope that these experiments sufficiently address the concerns raised about our description of JNK activity levels using the TRE-RFP reporter.

6) The idea that JNK impinges on the cell cycle seems plausible. The study evokes Cdc25 downregulation through Tribbles upregulation as a potential mechanism as how JNK signalling could impinge on the cell cycle. Surprisingly there is no mentioning of literature that JNK can phosphorylate Cdc25 in HEK293 cells attenuating its phosphatase activity (e.g. Gutierrez et al., 2010), therefore there is a direct link on Cdc25 activity regulation, without the need to downregulate Cdc25. The GFP trap line for string is apparently not homozygous viable and hence not functional, so I am not convinced that the data show that stg levels actually decline in G2 stalled cells.

Please see our response to point 4 as above:

Because of the literature on mammalian Cdc25c being phosphorylated by JNK to drive its proteasomal degradation (Goss et al., 2003; Gutierrez et al., 2010), we previously initiated studies to look at String protein levels using IF and Western blots, and to analyze String phosphorylation levels using Phospho-tag gel shift assays in different genetic backgrounds in vivo. However, we encountered several technical challenges: the *stg*-GFP trap does not give rise to a functional fusion protein, an HA-tagged Stg protein (FBrf0225451) is only detected in a very short window of the cell cycle (late G2/M) and thus hard to visualize on Western blots, and an antibody recognizing Stg in embryos (Di Talia et al., 2013) does not detect Stg in imaginal discs (possibly because of different fixation conditions). We thus have stopped this line of investigation until we have more resources to investigate Stg protein levels, modifications and activity in imaginal disc.

However, we included the references to the mammalian data (Goss et al., 2003; Gutierrez et al., 2010) now in the Discussion because we agree that these studies are important for our observations.

In addition, we agree that the stg-GFP trap line is a *stg* mutant and does not produce a functional protein. However, we argue that the *stg*-GFP trap line functions as an enhancer trap-like reporter of the *stg* locus allowing us to conclude that Stg is at least transcriptionally downregulated. We include the following data now in the main text and in Figure 4—figure supplement 1E-G’’:

“We first analyzed a GFP trap inserted in the *stg* locus (Buszczak et al., 2007). […] We observed that expression of the *stg*-GFP trap was dramatically downregulated in G2-shifted cells in *egr-*expressing discs (Figure 4A-B’)”.

7) JNK is tightly linked to the G2/M DNA damage response checkpoint. Can it be ruled out that this checkpoint is triggered in the conditions used? Pushing cells through the cell cycle by Cdc25 overexpression can override DNA damage response checkpoints in certain systems. It would be important to test that injury, temporal egr RNA and wts do not activate the DNA damage response (DDR). There are some important experiments in Supplementary Figure 7 in the manuscript in this regard, buy they are oddly placed and discussed. Is the reported G2 stalling clearly distinguishable from G2 cell cycle arrest induced by DNA damage?

Please see our response to point 4 above:

During the preparation of the manuscript we received feedback that describing the DNA damage data early on extends the first part of the manuscript at the expense of the important functional data for G2-stalling. As we absolutely agree with the reviewer that DNA damage is a crucial concept in cell cycle arrests, we have moved the data to Figure 3—figure supplement 1. We choose to present it in the supplements because our observations align with a previously published paper by the Johnston lab (Wells et al., 2006), who similarly observed that DNA damage is not involved in the regulation of a cell cycle arrest in undead cells (which we think represents the same phenomenon). We now emphasize in the text that JNK signaling is sufficient for G2-stalling but that it may be a downstream effector of cellular damage signals, such as heavy DNA damage or proteotoxic stress. Importantly, however, as JNK signaling during development (which would be independent of a (DNA) damage context) is sufficient for G2-stalling, we conclude that DNA damage could be an upstream regulator (via activation of JNK) but that itself is not necessary for stalling if other JNK-activating signals dominate.

Thus, we cannot and in fact we do not want to rule out that the DDR is activated. As irradiation also activates JNK in imaginal discs (for example (Pinal et al., 2018)), we suggests that DNA damage could be one of the many signals that drive JNK-dependent G2-stalling, however other cellular damage signals (proteotoxic damage, ER stress, lipid oxidation) may feed into JNK to stall the cell cycle as well.

8) While the non-autonomous activation of TRE is clear and also previously reported, the situation could be more complex. String overexpression might affect cell fitness and therefore cell competition could be an issue here. Can this be clearly separated, ruled out? That would be important to address.

As JNK signaling is involved in cell competition processes (for example (Kucinski et al., 2017), we have in fact been intrigued by how G2-stalling may affect competitive interactions in the tissue.

Currently, we do not have a clear answer to this question and suggest that new studies needs to further clarify questions related to the topic of cell competition. However, we offer these old and new experiments as a basis for a first discussion:

- Stg overexpression causes cells in injured discs to die (Figure 6A). This suggests that driving JNK-signaling cells through the cell cycle may cause them to become losers in this competition scenario.

- Stg overexpression in tumorigenic cells causes a reduction in non-autonomous overgrowth (Figure 7G-K’’). In a competition scenario, this could also occur through active killing of wild type cells by *wts^x1^, UAS-stg* or *dlg1^G0342^, UAS-stg* cells. In this scenario, driving JNK signaling cells through the cell cycle would make them super-competitors. (However, one would expect that the wild type size falls below that of wild type control discs, which is not the case.)

As these experiments would reach opposite conclusions about the winner or loser state of JNK-signaling G2-stalled cells, we had not discussed cell competition in the current form of the manuscript.

However, we wanted to more directly test if the lack of mitogenic signals or a gain of super-competitor status in *wts^x1^, UAS-stg* cellsprevents non-autonomous overgrowth of the surrounding wild type tissue.

1) To directly investigate if G2 stalled cells promote mitogenic signaling, we quantified mitotic counts using phospho-Histone 3 stainings in wild type tissue surrounding either *wts^x1^* or *wts^x1^, UAS-stg* clones (data now included in the manuscript in Figure 7L,M).

2) To investigate if G2 stalled engage in competitive interactions leading to killing of cells, we quantified the extent of apoptosis using Dcp-1 stainings in wild type tissue surrounding either *wts^x1^* or *wts^x1^, UAS-stg* clones (not included in the manuscript).

We found that the area fraction of apoptotic cells mildly increased in both the tumor and the surrounding wild type tissue by *UAS-stg* overexpression (0.14 ± 0.06 in *wts^x1^* vs 0.19 ± 0.05 in UAS stg, *wts^x1^*, and 0.06 ± 0.03 in wild type tissue surrounding *wts^x1^* vs 0.10 ± 0.04 in wild type tissue surrounding UAS stg, *wts^x1^*, n = 5 confocal section from each of n = 6 disc, for each data point). Because both wild type and tumorigenic cells responded with the same trend, it is hard to draw any conclusions about a change in competitive interactions by preventing G2-stalling which affects apoptosis in one population.

In contrast, we found that the rate of mitotic events in the wild type tissue is reduced to 50% by *UAS-stg* overexpression in *wts^x1^*cells. This strongly supports our conclusion that a reduction in mitogenic signals causes a reduction in non-autonomous overgrowth, when G2-stalling is impaired.

Combined, we propose that our data more directly supports a model where G2-stalling promotes non-autonomous mitogenic signals rather than altering competitive interactions in the tissue.

9) A more general concern is that novel bits seemed investigated in less detail than the already well established findings. The non-cell autonomous influence of tumour cells on neighbours to direct faster growth is fascinating but not well explored, and the suggestion that stalled cells were transiently senescent and releasing SARPs is too rushed to be definitive. Nonetheless, this is highlighted in the title and Abstract of this paper. Finally, are G2-stalled cells really protected from apoptosis? It seems that both pro- and anti-apoptotic pathways are activated; is knockdown of Hippo/Yorkie signalling in these cells needed?

Indeed, the mechanisms of how tumour cells influence neighbours to direct overgrowth are only beginning to be understood. Some of the non-autonomous signals implicated in the literature are Wg, Dpp, and JAK/STAT ligands as well as activation of Hippo/Yki signaling (reviewed for example in Enomoto et al., 2015). This aligns with our observation that *upd-*lacZ (JAK/STAT ligand) and *diap1-3.5*-GFP (Hippo/Yki signaling) are upregulated in *egr-*expressing and thus JNK-signaling and G2 stalled cells.

Many studies also link activation of initiator caspases to non-autonomous mitogenic signals. ‘Undead cells’, i.e. JNK-signaling cells that are resistant to apoptosis, uncover stimulation of the initiator caspases, such as Dronc, as a source for mitogenic signals that cause overgrowth on chronic timescales. On physiological timescales, Dronc activity functions in directing compensatory proliferation (for example (Chen, 2012; Fuchs and Steller, 2015; Martin et al., 2009; Perez-Garijo et al., 2009; Shlevkov and Morata, 2012; Wells et al., 2006)). These studies link activation of JNK and of initiator caspases, as well as resistance to apoptosis, to physiological and aberrant induction of mitogenic signals.

To better address how these studies fit our observations, we performed the following experiments:

1) To more directly investigate if G2 stalled cells promote proliferative signaling, we quantified mitotic counts using phospho-Histone 3 stainings in wild type tissue surrounding either *wts^x1^* or *wts^x1^, UAS-stg* clones (now included in the manuscript in Figure 7L,M). We found that the rate of mitotic events in the wild type tissue is reduced to 50% by *UAS-stg* overexpression *wts^x1^*cells. This strongly supports our conclusion that a reduction in mitogenic signals causes a reduction in non-autonomous overgrowth, when G2-stalling is impaired.

2) To investigate if ‘undead cells’ – an established model for non-autonomous overgrowth – are stalled in G2, we generated undead cells using our models by co-expressing p35 in *egr-*expressing disc. Importantly, these cells display a complete shift towards G2 (Figure 7N), suggesting that undead cells also arrest in G2. Thus G2-stalling is intimately associated with cellular states known to chronically stimulate growth.

3) To demonstrate that caspases are activated in G2 stalled cells and that G2 stalling may confer resistance to the execution of apoptosis, we analyzed the CasExpress sensor in surgically injured discs. CasExpress permanently labels anastatic cells, which have survived caspase activation (Ding et al., 2016; Tang et al., 2012). Strikingly, 24 h after surgical injury, we observed many CasExpress-positive clones near the wound site (Figure 7—figure supplement 2A.). This indicates that many wound-proximal cells survive caspase activation and subsequently proliferate. While we cannot demonstrate that these anastatic cells corresponded to G2-stalled cells, we suggest that G2 stalling could facilitate transient mitogenic signaling of Dronc-positive, JNK-activated cells at wound sites.

Of course, we agree that further studies need to clarify the regulation of SASPs in this context. We do not explicitly include SASPs per se in the title but wanted to emphasize the parallels between senescence states and G2-stalling, which we have investigated, such as:

3) Cell cycle arrest

4) Cell size increase

5) Apoptosis resistance

6) Mitogenic signals to surrounding cells

7) Upregulation of MMPs, cytokine expression, unfolded protein response and ROS metabolism

These markers are hallmarks of mammalian senescent cells. We would thus like to keep senescence in the title but changed the title to “JNK-dependent cell cycle stalling in G2 promotes survival and senescence-like phenotypes in tissue stress”. We also removed ‘SASP’ from the Abstract.

With respect to Hippo/Yki signaling: Yki activation by JNK signaling in *egr-*expressing cells has been previously reported (Sun and Irvine, 2011). This study also describes that knock-down of *yki* in *egr-*expressing cells using an RNAi construct completely ablates this cell population and prevents successful regeneration (see Figure 3 in Sun and Irvine, (2011)). Thus, the same population that we identified to be G2-stalled cells depend on the potent anti-apoptotic factor Yki for survival.